# TURBO-DDCM: FAST AND FLEXIBLE ZERO-SHOT DIFFUSION-BASED IMAGE COMPRESSION

**Amit Vaisman**[1], **Guy Ohayon**[2], **Hila Manor**[1], **Michael Elad**[1], **Tomer Michaeli**[1]

[1]Technion – Israel Institute of Technology    [2]Flatiron Institute, Simons Foundation

`amit.vaisman@campus.technion.ac.il, gohayon@flatironinstitute.org,`
`{hila.manor@campus, elad@cs, tomer.m@ee}.technion.ac.il`

## ABSTRACT

While zero-shot diffusion-based compression methods have seen significant progress in recent years, they remain notoriously slow and computationally demanding. This paper presents an efficient zero-shot diffusion-based compression method that runs substantially faster than existing methods, while maintaining performance that is on par with the state-of-the-art techniques. Our method builds upon the recently proposed Denoising Diffusion Codebook Models (DDCMs) compression scheme. Specifically, DDCM compresses an image by sequentially choosing the diffusion noise vectors from reproducible random codebooks, guiding the denoiser's output to reconstruct the target image. We modify this framework with *Turbo-DDCM*, which efficiently combines a large number of noise vectors at each denoising step, thereby significantly reducing the number of required denoising operations. This modification is also coupled with an improved encoding protocol. Furthermore, we introduce two flexible variants of Turbo-DDCM, a priority-aware variant that prioritizes user-specified regions and a distortion-controlled variant that compresses an image based on a target PSNR rather than a target BPP. Comprehensive experiments position Turbo-DDCM as a compelling, practical, and flexible image compression scheme. Code is available on our project's webpage.

## 1 INTRODUCTION

The field of image compression has witnessed a shift towards neural-based approaches in recent years (Ballé et al., 2017; Toderici et al., 2017; Li & Ji, 2020), and more recently to methods that rely on diffusion models (Sohl-Dickstein et al., 2015; Ho et al., 2020; Song et al., 2020). In particular, diffusion models have been utilized for image compression by training dedicated models (Ghouse et al., 2023; Yang & Mandt, 2023), by fine-tuning existing models (Careil et al., 2024), or by using them in a zero-shot manner (Theis et al., 2022; Elata et al., 2024; Ohayon et al., 2025; Vonderfecht & Liu, 2025). In principle, zero-shot methods are appealing because they enable the same diffusion backbone to be shared across multiple tasks (such as compression, restoration, editing, generation, etc.), thereby allowing all tasks to benefit simultaneously from improvements to the shared backbone. However, existing zero-shot diffusion compression methods remain impractical due to their high computational demands, which result in slow inference. Ranging from approximately 10 seconds (achieved using a custom CUDA kernel (Vonderfecht & Liu, 2025)) to several minutes (Elata et al., 2024) for compressing and decompressing a single image, these slow inference times make zero-shot diffusion compression methods less compelling compared to training-based methods that operate much faster.

This work presents Turbo-DDCM, a fast and practical zero-shot diffusion-based compression method. It achieves a round-trip compression-decompression time of 1.8 seconds per image without custom hardware-specific optimizations, while maintaining competitive performance to state-of-the-art approaches. Our method extends Denoising Diffusion Codebook Model (DDCM) (Ohayon et al., 2025), a recently proposed zero-shot diffusion-based compression method that demonstrated state-of-the-art results. Specifically, DDCM compresses a target image by carefully selecting noise vectors during the generative diffusion process, in a manner that minimizes the denoising error between the target image and the denoiser's output at each step. Importantly, each noise vector is picked

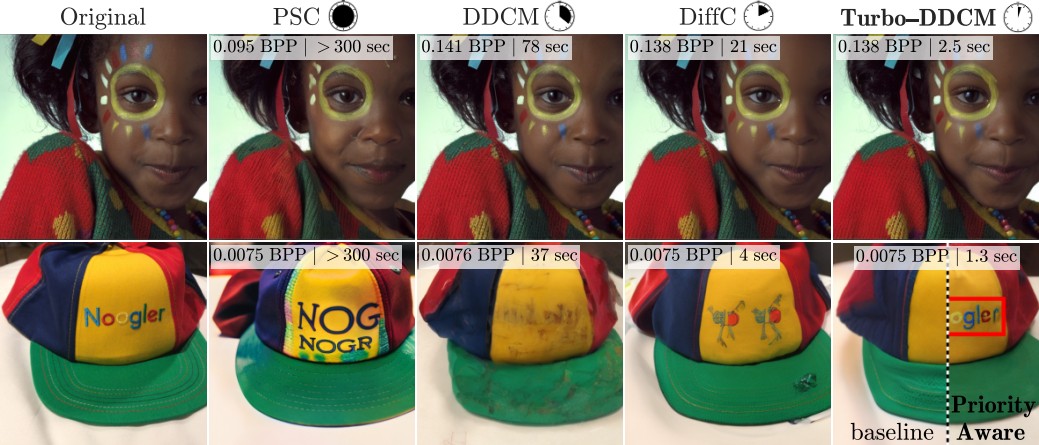

Figure 1: **Turbo-DDCM:** Our method provides reconstructions with equal or better fidelity compared to previous methods, while being much faster. At the same BPP and runtime, the priority-aware variant (bottom-right) better serves key regions of choice.

from a reproducible Gaussian codebook, implying that the final generated image can be efficiently stored/transmitted by storing/transmitting the indices of the selected noise vectors. While such a simple compression mechanism works surprisingly well, it requires hundreds of denoising steps to achieve sufficient reconstruction quality, which results in an average round-trip compression-decompression time of 65 seconds. Our proposed Turbo-DDCM compression method accelerates this process by significantly reducing the required number of denoising steps to just a few dozen. In particular, Turbo-DDCM combines a large number of noise vectors at each DDCM step by solving a sparse least-squares optimization problem in closed form, leveraging the near-orthogonality of Gaussian codebook vectors in high-dimensional spaces. While Ohayon et al. (2025) proposed a seemingly similar approach that utilizes a matching pursuit (MP) strategy (Mallat & Zhang, 1993), their method relies on a greedy iterative process based on convex combinations and an exhaustive search. As a result, the scalability of this approach is limited due to runtime constraints. In contrast, our solution for combining codebook noise vectors is significantly faster and more computationally efficient. Specifically, it achieves orders of magnitude faster runtime compared to the method proposed by Ohayon et al. (2025), while producing equivalent results (see App. D for theoretical justifications and empirical evaluations). Indeed, we show that the high scalability of our noise combination strategy enables a 92% reduction in the number of required diffusion steps. Beyond efficiency, we also propose a new bit-coding protocol for effective encoding of the indices of the selected noise vectors. These contributions establish Turbo-DDCM as competitive with existing zero-shot methods in terms of reconstruction quality, while achieving dramatically lower runtime (see example in Fig. 1).

Finally, we introduce two flexible variants of Turbo-DDCM. The first supports priority-aware compression (Li et al., 2023; Xu et al., 2025), which allows allocating more bits to arbitrarily shaped user-specified regions in the target image to improve reconstruction quality in those regions (see Fig. 1 for an example). The second variant targets a user-specified PSNR rather than a fixed bitrate. This is useful since, at a fixed bitrate, Turbo-DDCM and alternative zero-shot methods yield highly variable distortion across images. To the best of our knowledge, our work is the first to incorporate such capabilities into a zero-shot diffusion-based compression method.

To summarize, this paper introduces Turbo-DDCM – a novel and highly efficient diffusion-based image compression algorithm, with the following features:

- *Zero-shot:* Turbo-DDCM relies on a pre-trained latent diffusion generator, without any need for further training or fine-tuning. As such, the backbone diffusion model can be replaced flexibly to allow improved future versions.
- *Performance:* Turbo-DDCM has a competitive performance with the current state-of-the-art methods in terms of the rate-distortion-perception tradeoff.

- *Speed:* Turbo-DDCM is the fastest zero-shot method, achieving nearly an order of magnitude speedup over the fastest existing approach. Our method operates without any custom hardware-specific acceleration, maintaining nearly constant runtime across bitrates. This is achieved via a better noise sampling mechanism, which in turn allows using a shallower diffusion process. Coupled with a new bitstream protocol, performance is not compromised.
- *Perfect bitrate control:* Our method provides a predictable and constant bitrate across images, which can be finely controlled via a single hyperparameter across a wide range of bitrates.
- *Priority-aware:* We introduce a variant of Turbo-DDCM that enables enhanced reconstruction fidelity in user-selected regions of the image, with controllable prioritization levels.
- *Distortion control:* We present a distortion-geared Turbo-DDCM variant that targets a specific PSNR for each image (instead of target bitrate), addressing the variable distortion in zero-shot methods at fixed bitrate.

## 2 RELATED WORK

**Non-Zero-Shot Diffusion-based Image Compression.** Recent advances in diffusion-based compression have demonstrated impressive rate-distortion-perception performance by training models from scratch (Yang & Mandt, 2023; Ghouse et al., 2023; Iwai et al., 2024) or fine-tuning existing diffusion models (Körber et al., 2024; Careil et al., 2024). More recently, several one-step methods (Park et al., 2025; Xue et al., 2025) have tried to bypass the computational cost of iterative denoising by directly learning a mapping from the latent code to the clean signal in a single reverse step. However, all these approaches share the drawback of requiring training or fine-tuning tailored for compression. In contrast, zero-shot methods preserve the diffusion model as a multi-purpose backbone, which can also serve for generation (Ho et al., 2020; Song et al., 2020), restoration (Kadkhodaie & Simoncelli, 2021; Kawar et al., 2022; Raphaeli et al., 2025; Man et al., 2025), editing (Manor & Michaeli, 2024; Cohen et al., 2024), etc.

**Zero-shot Diffusion-based Image Compression.** Rather than training models or fine-tuning existing models, some recent methods use pretrained diffusion models in a zero-shot manner for image compression. IPIC (Xu et al., 2024) adopts a compression method based on posterior-sampling. PSC (Elata et al., 2024) and DiffC (Theis et al., 2022) harness ideas from compressed sensing (Donoho, 2006) and reverse channel coding (RCC) (Theis & Yosri, 2022), respectively. DDCM (Ohayon et al., 2025) changes the standard DDPM process (Ho et al., 2020) by sampling from a quantized Gaussian space, offering a simpler approach. However, all existing methods suffer from prohibitive computational demands, often requiring hundreds or even thousands of diffusion steps to compress a single image, making them unsuitable for practical usage. Recently, the RCC protocol used by DiffC was implemented on an optimized CUDA kernel (Vonderfecht & Liu, 2025), which alleviates much of the computational demands. However, this method remains limited by custom hardware-specific accelerations, large deviations from target bitrate across different input images, and substantial runtime variation across different compression bitrates. In contrast, our proposed method is the fastest by a wide margin, without custom hardware-dependent optimizations, nearly constant runtime across all bitrates and a constant bitrate between different images given the same target bitrate. At the same time, DiffC, DDCM, and our Turbo-DDCM share fundamental concepts in their underlying design, but extend it in different directions, as detailed in App. H.

**ROI (priority-aware) compression methods.** Region-of-interest (ROI) compression (Li et al., 2023; Jin et al., 2025) prioritizes user-specified regions of an image by allocating to them a larger portion of the bits, resulting in higher fidelity in those regions on the expense of others. This paradigm can be useful in medical imaging (Srivastava & Fujii, 2025), video conference calls and more. Recently, it was demonstrated in diffusion-based compression (Xu et al., 2025). To the best of our knowledge, we are the first to apply ROI compression to a zero-shot diffusion method. We do so in a general way, enabling per-pixel prioritization, which we refer to as priority-aware compression.

Concurrent with our work, Su & Kasai (2025) investigate one of our main ideas, multiple choice from a reproducible codebook, primarily in the context of inverse problems.

## 3 BACKGROUND

### 3.1 DENOISING DIFFUSION PROBABILISTIC MODELS (DDPM)

Diffusion models (Sohl-Dickstein et al., 2015; Ho et al., 2020; Song et al., 2020) generate samples from a data distribution $p_0$ by learning to reverse a forward diffusion process. Specifically, for $t \in \{1, \ldots, T\}$, the forward process gradually corrupts the data $\mathbf{x}_0 \sim p_0$ with noise via

$$\mathbf{x}_t = \sqrt{\alpha_t}\mathbf{x}_{t-1} + \sqrt{1 - \alpha_t}\boldsymbol{\epsilon}_t, \quad \boldsymbol{\epsilon}_t \sim \mathcal{N}(\mathbf{0}, \boldsymbol{I}), \tag{1}$$

where $\alpha_1, \ldots, \alpha_T > 0$ are time-dependent constants controlling the signal-to-noise ratio. In DDPM (Ho et al., 2020), the reverse diffusion process generates samples from the data distribution by gradually denoising a random noise sample $\mathbf{x}_T \sim \mathcal{N}(\mathbf{0}, \boldsymbol{I})$ via

$$\mathbf{x}_{t-1} = \boldsymbol{\mu}_t(\mathbf{x}_t) + \sigma_t \mathbf{z}_t, \quad \mathbf{z}_t \sim \mathcal{N}(\mathbf{0}, \boldsymbol{I}), \tag{2}$$

where $\sigma_t = \sqrt{1 - \alpha_t}$, and $\boldsymbol{\mu}_t(\mathbf{x}_t)$ is the conditional mean of $\mathbf{x}_{t-1}$ given $\mathbf{x}_t$. $\boldsymbol{\mu}_t(\mathbf{x}_t)$ can be expressed in terms of the minimum mean-squared-error (MMSE) estimator $\hat{\mathbf{x}}_{0|t}$ of $\mathbf{x}_0$ given $\mathbf{x}_t$,

$$\boldsymbol{\mu}_t(\mathbf{x}_t) = \frac{\sqrt{\bar{\alpha}_{t-1}}\,(1 - \alpha_t)}{1 - \bar{\alpha}_t}\hat{\mathbf{x}}_{0|t} + \frac{\sqrt{\alpha_t}\,(1 - \bar{\alpha}_{t-1})}{1 - \bar{\alpha}_t}\mathbf{x}_t, \tag{3}$$

where $\bar{\alpha} = \prod_{s=1}^{t} \alpha_s$. The MMSE estimator $\hat{\mathbf{x}}_{0|t}$ plays a central role in the rest of this paper.

### 3.2 DENOISING DIFFUSION CODEBOOK MODELS (DDCM)

DDCM (Ohayon et al., 2025) modifies the reverse process of DDPM (eq. (2)) by replacing the random Gaussian noise sampling with a selection of noises from reproducible codebooks $\boldsymbol{C}_t$. Specifically, each codebook $\boldsymbol{C}_t$ is an ordered set filled with $K$ i.i.d. white Gaussian noise vectors, which we refer to as *atoms*,

$$\mathcal{C}_t = \left[ \boldsymbol{z}_t^{(1)}, \boldsymbol{z}_t^{(2)}, \ldots, \boldsymbol{z}_t^{(K)} \right], \quad t = 2, \ldots, T + 1, \tag{4}$$

where, as in DDPM, no noise is added at $t = 1$. The DDCM generation step then becomes

$$\mathbf{x}_{t-1} = \boldsymbol{\mu}_t(\mathbf{x}_t) + \sigma_t \boldsymbol{C}_t(\mathbf{k}_t), \tag{5}$$

where $\mathbf{k}_t \overset{\text{i.i.d.}}{\sim} \text{Unif}(\{1, \ldots, K\})$. This modification yields a discrete yet highly expressive generated distribution, as the number of possible distinct output samples grows exponentially with the number of diffusion steps.

The primary property of DDCM is its ability to act as a zero-shot image compression algorithm, achieved by creating all the codebooks once and keeping them constant for all subsequent applications of the model (e.g., by using a shared random seed). Specifically, a target image is compressed by selecting, at each diffusion step, the fixed codebook atom that best matches the target image. Formally, to compress a target image $\mathbf{x}_0$, Ohayon et al. (2025) compute the denoising residual between $\mathbf{x}_0$ and the MMSE estimation $\hat{\mathbf{x}}_{0|t}$ at each timestep $t$, and select the codebook entry that maximizes the inner product with this residual:

$$k_t = \underset{k \in \{1, \ldots, K\}}{\arg\max} \langle \boldsymbol{C}_t(k), \mathbf{x}_0 - \hat{\mathbf{x}}_{0|t} \rangle, \tag{6}$$

where $k_t$ is the selected codebook entry at timestep $t$. This sampling process results in an ordered sequence of chosen indices $(k_t)_{t=2}^{T}$, whose binary representation serves as the compressed form of the image. Decompression then follows eq. (5), where instead of randomly sampling from the codebooks, the stored indices are re-selected deterministically. Consequently, the bits-per-pixel (BPP) of DDCM can easily be computed via

$$\text{BPP}_{\text{DDCM}} = \frac{(T - 1)\lceil \log_2(K) \rceil}{\text{number of pixels}}. \tag{7}$$

While this strategy leads to good results, it is practically limited only to the regime of extremely low bitrates. Indeed, even when using $T = 1000$ diffusion steps and codebooks with $K = 16{,}384$ atoms,

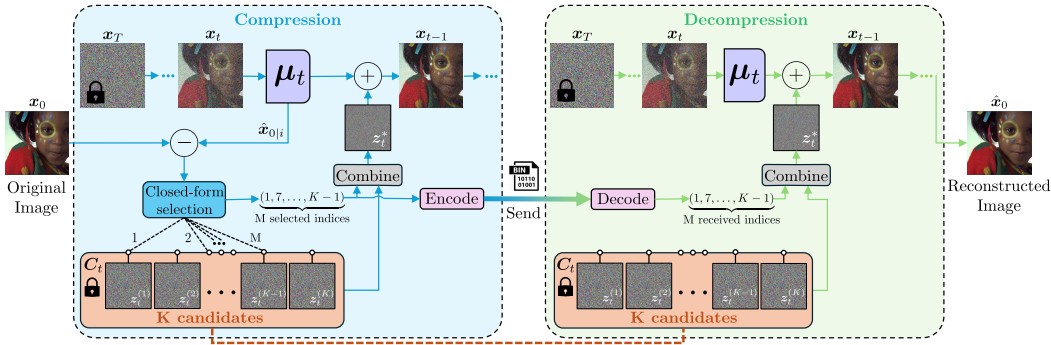

Figure 2: **Turbo-DDCM overview:** Building on DDCM, we replace its random noise sampling with an effective and efficient closed-form selection rule that can quickly combine an arbitrary number of noise vectors, enabling significantly fewer diffusion steps. The selected indices are encoded using our new bit transmission protocol, which achieves substantially higher encoding efficiency than DDCM's protocol. The decoder reconstructs the image by running the generative diffusion process while re-selecting the codebook noise vectors that correspond to the decoded indices. This results in a zero-shot compression method that is both highly efficient and competitive in performance.

the bitrate for a $768 \times 768$ image is only about 0.024 BPP. Increasing $T$ applies the denoiser more times, which is computationally expensive, while increasing $K$ increases the bitrate logarithmically but makes searching for the correct noise much more demanding. To enable higher bitrates, the authors propose a refinement strategy for noise selection based on matching pursuit (MP) (Mallat & Zhang, 1993). Specifically, at each step $t$, the chosen noise is constructed as a convex combination of $M$ elements from $C_t$, selected greedily to maximize correlation with the residual $\mathbf{x}_0 - \hat{\mathbf{x}}_{0|i}$ (as in eq. (6)). This combination involves $M - 1$ quantized scalar coefficients, each drawn from a set of $2^C$ values within $[0, 1]$ (where C is the number of bits needed to communicate each coefficient). Transmitting the $M$ selected noise indices requires $\lceil \log_2(K) \rceil M$ bits per timestep, along with $C(M - 1)$ bits for the quantized coefficients. When using this approach, the BPP becomes

$$\text{BPP}_{\text{DDCM with MP}} = \frac{(T - 1)\big(\lceil \log_2(K) \rceil M + C(M - 1)\big)}{\text{number of pixels}}. \tag{8}$$

This MP strategy widens the bitrate range of DDCM. In practice, however, it incurs extremely high runtime due to its iterative nature, as one MP process can take more than 0.1 seconds and must be performed after each diffusion step, resulting in a total cost multiplied by $T$. Moreover, due to its greedy design, increasing $M$ alone provides only limited gains in correlation with the residual; $C$ must also be increased, while runtime grows exponentially with $C$, which further limits scalability. See Apps. C and D for theoretical justifications and empirical demonstrations.

## 4 METHOD

We now turn to introduce Turbo-DDCM. This includes a new and highly efficient approach for combining *many* codebook atoms at each of DDCM's denoising steps. Our approach yields additional advantages across multiple aspects of compression beyond the acceleration of DDCM's matching pursuit. We complement our new noise construction method with a novel encoding protocol. An overview of our approach is shown in Fig. 2 and pseudo-code is provided in App. I.

### 4.1 EFFICIENT MULTI-ATOM SELECTION

We approximate the residual vector $\mathbf{x}_0 - \hat{\mathbf{x}}_{0|t}$ with a linear combination of exactly $M$ atoms having nonzero quantized coefficients from a fixed ordered set $\mathcal{V}$. Formally, our optimization problem for timestep $t$ is a constrained least squares of the form

$$\mathbf{s}_t^* = \underset{\mathbf{s}_t \in \mathbb{R}^K}{\arg\min} \left\| C_t \mathbf{s}_t - (\mathbf{x}_0 - \hat{\mathbf{x}}_{0|t}) \right\|_2^2 \quad \text{s.t. } \|\mathbf{s}_t\|_0 = M, \ \forall i \, (\mathbf{s}_t)_i \in \mathcal{V} \cup \{0\}, \tag{9}$$

where $(\mathbf{s}_t)_i$ denotes the $i$-th entry of the vector $\mathbf{s}_t$. Using the solution $\mathbf{s}_t^*$, we define

$$\mathbf{z}_t^* = \frac{\boldsymbol{C}_t \mathbf{s}_t^*}{\mathrm{std}(\boldsymbol{C}_t \mathbf{s}_t^*)}. \tag{10}$$

Finally, the Turbo-DDCM sampling process is defined as

$$\mathbf{x}_{t-1} = \boldsymbol{\mu}_t(\boldsymbol{x}_t) + \sigma_t \mathbf{z}_t^*. \tag{11}$$

To solve the optimization problem in eq. (9), we use an efficient closed-form solution. The key insight of our approach is that Gaussian random codebooks are nearly orthogonal in high-dimensional spaces (see App. B for how this approximate orthogonality affects performance). Under this assumption, the thresholding algorithm (Elad, 2010) provides a closed-form solution for sparse least squares (eq. (9) without the quantization constraint):

$$(\mathbf{s}_t^*)_i = \begin{cases} (\mathbf{u}_t)_i / \|\mathbf{z}_t^{(i)}\|_2, & i \in \mathrm{TopM}(|\mathbf{u}_t|) \\ 0, & \text{otherwise} \end{cases}, \quad (\mathbf{u}_t)_i = \frac{\langle \mathbf{z}_t^{(i)}, \mathbf{x}_0 - \hat{\mathbf{x}}_{0|t} \rangle}{\|\mathbf{z}_t^{(i)}\|_2}, \; i = 1, \dots, K, \tag{12}$$

where $\mathbf{z}_t^{(i)}$ is the $i$-th column of $\boldsymbol{C}_t$ and $\mathrm{TopM}(|\mathbf{u}|)$ denotes the indices of the $M$ largest entries of $|\mathbf{u}|$. We extend this solution to incorporate the quantization constraint. We then adapt it to our specific setting of Gaussian i.i.d. codebooks and our characteristic configurations which share all $\mathcal{V} = [-1, +1]$, resulting in the following algorithm:

$$(\mathbf{s}_t^*)_i = \begin{cases} \mathrm{sign}((\mathbf{u}_t)_i), & i \in \mathrm{TopM}(|\mathbf{u}_t|) \\ 0, & \text{otherwise} \end{cases}, \quad (\mathbf{u}_t)_i = \langle \mathbf{z}_t^{(i)}, \mathbf{x}_0 - \hat{\mathbf{x}}_{0|t} \rangle, \quad i = 1, \dots, K. \tag{13}$$

See App. C for the derivation and App. E for the rationale behind the chosen $\mathcal{V}$.

Our approach fundamentally differs from DDCM's MP strategy in four key aspects, leading to enhanced capabilities and desirable properties.

**Noise Construction Efficiency.** Our closed-form algorithm is orders of magnitude faster than DDCM MP, which relies on an iterative search with exhaustive evaluations. Consequently, we construct the noise at each diffusion step much more efficiently.

**Number of Required Diffusion Steps.** Unlike DDCM, our method maintains a nearly constant runtime as $M$ increases (App. D). Thus, we can scale $M$ to hundreds, whereas DDCM MP is practically limited to very small values. This enables a major reduction in diffusion steps, as selecting many atoms per step leads to stronger residual estimations. Consequently, a few diffusion steps with strong estimations can replace many steps with weak ones. Overall, the number of diffusion steps is reduced by more than 92% for comparable compression quality.

**Possible Quantization Levels.** While DDCM's MP is restricted to non-negative quantization levels due to its reliance on convex combinations, our approach allows both positive and negative coefficients. This effectively doubles the representational capacity, as allowing negative coefficients enables pointing in opposite directions, thereby increasing the number of directions in latent space available to approximate the residual and improving the solution to the optimization problem.

**Hyperparameters.** Both DDCM and Turbo-DDCM rely on four hyperparameters that influence bitrate: $T$, $K$, $M$, and $C$. Controlling bitrate via $T$ increases expensive denoiser calls; varying $K$ slows optimization in both methods; and varying $C$ alone yields only marginal gains. In our thresholding-based approach, increasing $M$ enhances representational power with negligible runtime cost and allows fine-grained bitrate control. Consequently, Turbo-DDCM can adjust bitrate using $M$ alone, ensuring runtime stability across bitrates and avoiding inefficient hyperparameter combinations. DDCM, however, cannot benefit from increasing $M$ in isolation and must simultaneously increase $C$. As it is not computationally efficient in both, it must also increase $K$ and $T$, resulting in significant runtime variation across bitrates. Moreover, DDCM is highly sensitive to inefficient hyperparameter combinations, which can further degrade performance. Detailed justifications and demonstrations are provided in Apps. C, D and E.

Due to the large reduction in $T$ described above, we replace the synthesized noise $\mathbf{z}_t^*$ in the final steps with DDIM sampling to improve perceptual quality at low bitrates. The number of DDIM

steps, $N$, is determined heuristically and decreases with increasing bitrate. Specifically, for a $T$-steps scheduler, the encoder transmits $\mathbf{z}_t^*$ for the first $T - N$ steps, which the decoder reconstructs, followed by $N$ DDIM steps executed only at the decoder. See App. A for exact details.

## 4.2 Efficient Bit Protocol for Large-M Combinations

When large $M$ values are employed, using the DDCM's bit protocol leads to poor compression due to redundancies. Specifically, DDCM encodes atom selections with $\lceil \log_2(K) \rceil M$ bits per diffusion step, preserving the order in which the atoms were selected, which is crucial for the noise constriction on the decoder. However, in our thresholding-based approach, atom ordering is semantically meaningless and only identity matters. Thus, a naive encoding of $M$ indices creates $M!$ equivalent representations per step, yielding $(M!)^{T-1}$ identical compressed representations. For extremely modest parameters of $M = 5$ and $T = 30$, this results in $120^{29} \approx 2^{200}$ equivalent representations, underscoring the need for a more efficient encoding protocol.

Each diffusion step requires communicating a combination of $M$ indices from a codebook of size $K$ (without repetition and order-invariant) along with quantized coefficients ($C$ bits each). The number of distinct choices is $\binom{K}{M} \cdot (2^C)^M$, requiring

$$\left\lceil \log_2 \left( \binom{K}{M} \cdot (2^C)^M \right) \right\rceil = \left\lceil \log_2 \left( \binom{K}{M} \right) \right\rceil + MC \tag{14}$$

bits per step. This gives the worst-case lower bound on the bits needed to encode this information.

We propose an encoding protocol that achieves this bound. Our protocol transmits the lexicographical index of the selected $M$-atom combination from within the set $\left\{ 1, \ldots, \binom{K}{M} \right\}$ of possible indices, followed by the selected $M$ quantized coefficients in canonical order. This approach eliminates the observed factorial redundancy while transmitting effectively the same information. Consequently, Turbo-DDCM's BPP is

$$\mathrm{BPP}_{\text{Turbo-DDCM}} = \frac{(T - N - 1) \left( \left\lceil \log_2 \left( \binom{K}{M} \right) \right\rceil + MC \right)}{\text{number of pixels}}. \tag{15}$$

As shown in App. F, our protocol reduces BPP by $\sim 40\%$ for typical Turbo-DDCM configurations.

## 5 Experiments

We evaluate our compression method on the Kodak24 (Franzen, 1999) and DIV2K (Agustsson & Timofte, 2017) datasets, center-cropping to $512 \times 512$. We compare to existing zero-shot diffusion-based methods, including PSC (Elata et al., 2024), DDCM (Ohayon et al., 2025), and DiffC (Theis et al., 2022), using the custom CUDA kernel implementation of DiffC (Vonderfecht & Liu, 2025). Additionally, we compare to non-neural, fine-tuning-based and training-based approaches, including BPG (Bellard, 2018), PerCo (SD) (Körber et al., 2024; Careil et al., 2024), ILLM (Muckley et al., 2023), two CRDR (Iwai et al., 2024) configurations, HiFiC (Mentzer et al., 2020), DiffEIC (Li et al., 2025) and StableCodec (Zhang et al., 2025). All zero-shot methods use the same pre-trained diffusion model, SD 2.1 Base. For Turbo-DDCM we use $T = 30$, $K = 16{,}384$, $C = 1$ and vary $M$ from 45 to 300 to control the bitrate. Distortion is measured using PSNR and LPIPS (Zhang et al., 2018), while perceptual quality is evaluated with FID (Bińkowski et al., 2018), computed on $64 \times 64$ patches following Mentzer et al. (2020). Runtime is measured based on process time for a round-trip compression-decompression on an NVIDIA A40 GPU.

**Compression Quality.** As shown in Figs. 3 and 4, Turbo-DDCM achieves competitive results against both zero-shot and non-zero-shot methods. Among zero-shot approaches, Turbo-DDCM surpasses PSC both in terms of distortion and perceptual quality. Compared to DDCM, Turbo-DDCM achieves equal or better distortion and perceptual quality. At low bitrates, DiffC yields better perceptual quality than Turbo-DDCM, with equal distortion. At high bitrates, DiffC and Turbo-DDCM present nearly the same distortion and perceptual quality. Compared to the other methods (non-neural, fine-tuning-based and training-based approaches), Turbo-DDCM surpasses all prior

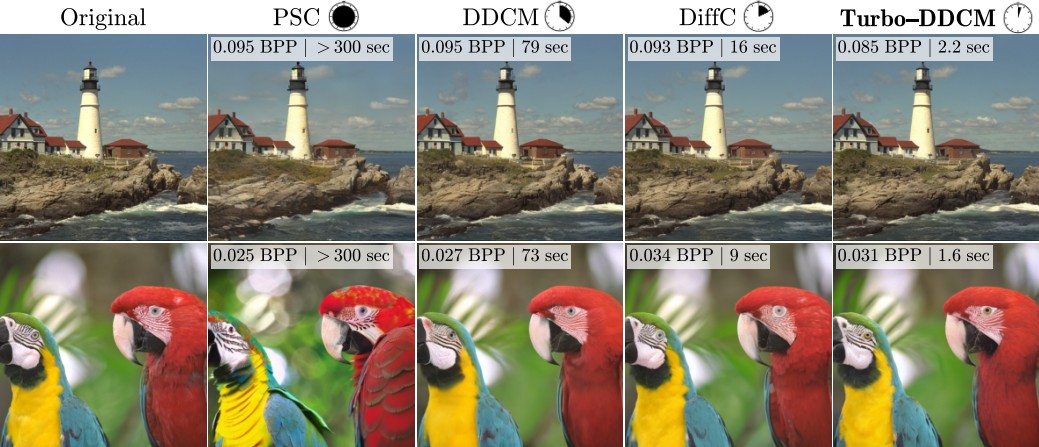

Figure 3: **Qualitative results:** The presented images are taken from the Kodak24 ($512 \times 512$) dataset. Our method produces highly realistic reconstructions while achieving over a $5\times$ speedup compared to previous approaches, depending on the bitrate.

methods on the rate–distortion–perception plane (Blau & Michaeli, 2019), except for StableCodec, which is specialized for each individual bitrate. Yet, at low bitrates, Turbo-DDCM still achieves better distortion compared to other perceptual-quality-oriented approaches, such as PerCo (SD) and DiffEIC, despite being a zero-shot method. However, for high bitrates, our method underperforms in distortion due to the encoder-decoder distortion bound imposed by SD 2.1. When using a latent diffusion model such as SD, compression is applied in the latent representation space rather than in the image pixel space. Since the encoder–decoder already introduces distortion, the maximum attainable quality of any latent-based method is bounded by this reconstruction error, forming a natural upper limit (Körber et al., 2024; Elata et al., 2024). Accordingly, Fig. 4 reports this bound. See App. A for additional evaluations.

**Runtime Performance.** As shown in Figs. 3 and 4, among zero-shot methods, Turbo-DDCM achieves state-of-the-art runtime by a significant margin. It outperforms DDCM by more than an order of magnitude, achieving over $34\times$ speedup at high bitrates. Turbo-DDCM also outperforms DiffC, with a $3\times$ acceleration at low bitrates and nearly an order of magnitude at high bitrates. Importantly, this performance advantage is achieved even though DiffC employs a custom CUDA kernel. As runtime might be implementation-dependent, we also compare the number of neural-function-evaluations (NFEs), which correspond to denoiser activations in this case. Even though NFE is implementation independent, it is imperfect, since it ignores operations performed between denoiser activations, which might also be computationally demanding. Nevertheless, the trends and differences in NFEs closely follow those observed for runtime, as detailed in App. A, along with additional evaluations. Among other methods, Turbo-DDCM is faster than HiFiC, DiffEIC, CRDR, and PerCo (SD). However, it is slower than BPG (non-neural), ILLM (non-diffusion-based), and StableCodec (non-zero-shot).

## 6 TURBO-DDCM VARIANTS

While traditional diffusion-based methods allocate bits uniformly across the image, some applications (e.g. medical imaging) benefit from prioritizing important regions. This paradigm, commonly referred to as ROI (region-of-interest) compression (Li et al., 2023; Srivastava & Fujii, 2025), improves fidelity in specified areas at the expense of others by non-uniform bit allocation. We introduce a variant of Turbo-DDCM that supports ROI compression through per-pixel prioritization, which we term priority-aware compression.

Our adaptation is based on a latent priority mask $\boldsymbol{w} \in \mathbb{R}_+^d$, where each entry specifies the relative weight of the corresponding latent coordinate, obtained by down-sampling a pixel-space prioritiza-

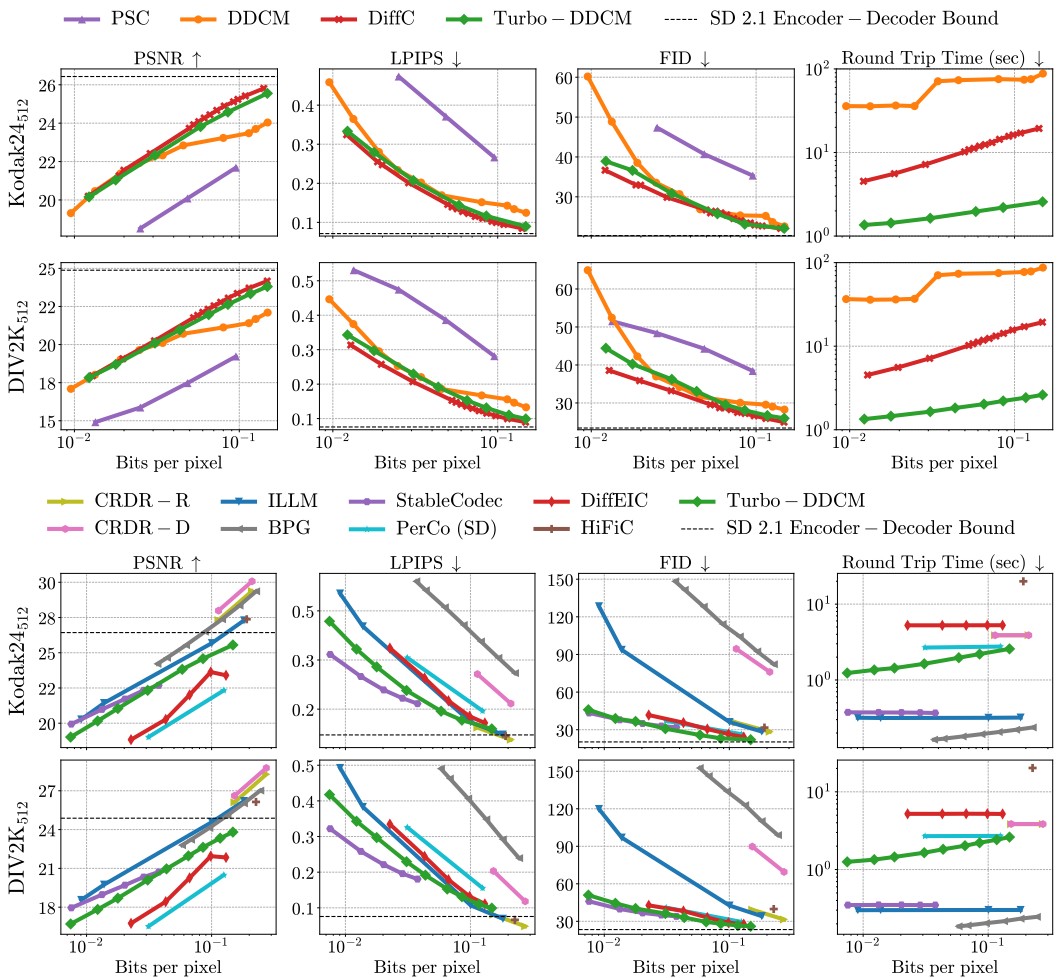

Figure 4: **Quantitative evaluation of rate-distortion-perception and runtime performance:** We compare Turbo-DDCM against zero-shot diffusion-based compression methods (top two rows) and other methods (bottom two rows), by evaluating distortion (PSNR or LPIPS), perceptual quality (FID), and runtime (round-trip compression–decompression time, in seconds). PSC's runtime is omitted due to its extreme complexity (>300 seconds/image). Overall, Turbo-DDCM achieves superior or competitive rate-distortion-perception performance compared to previous zero-shot methods, but with a significantly faster runtime (nearly an order of magnitude faster or more at high bitrates). The dashed vertical line in each subplot corresponds to the encoder-decoder distortion bound imposed by SD 2.1, attained by passing the clean images through this encoder-decoder without any compression. Since all the compared zero-shot methods and PerCo (SD) rely on SD 2.1, they all suffer from this distortion bound.

tion map. We then extend the optimization problem in eq. (9) to

$$\mathbf{s}_t^* = \underset{\mathbf{s}_t \in \mathbb{R}^K}{\arg\min} \ \|\boldsymbol{C}_t \mathbf{s}_t - \boldsymbol{w} \odot (\mathbf{x}_0 - \hat{\mathbf{x}}_{0|t})\|_2^2 \quad \text{s.t.} \quad \|\mathbf{s}_t\|_0 = M, \ \ \forall i \, (\mathbf{s}_t)_i \in \mathcal{V} \cup \{0\}. \tag{16}$$

Here, the residual vector $(\mathbf{x}_0 - \hat{\mathbf{x}}_{0|t})$ is weighted by $\boldsymbol{w}$, scaling each error according to its corresponding pixel priority. The $M$ codebook atoms and their quantized coefficients are then chosen accordingly. Importantly, $\boldsymbol{w}$ is not transmitted to the decoder, leaving both the encoding protocol and the BPP unchanged. Additionally, this modification has a negligible impact on runtime. As demonstrated in App. A, this adaptation generalizes naturally to both DDCM and DiffC methods.

We evaluate our variant on the Kodak24 ($512 \times 512$) dataset and compare to the other zero-shot methods, including the baseline Turbo-DDCM. Figure 5 shows that, unlike regular zero-shot methods,

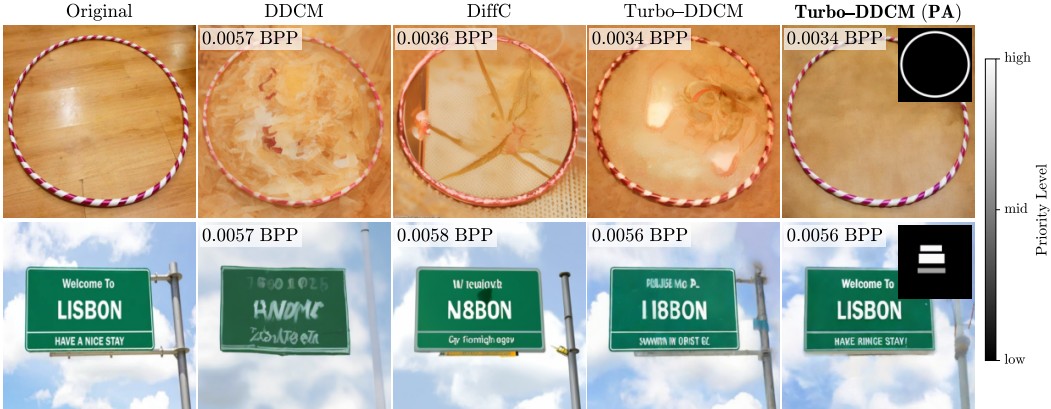

Figure 5: **Qualitative results of the priority-aware (PA) variant:** Regular methods fail to reconstruct key regions, whereas our PA variant reconstructs them faithfully according to the prioritization mask. In the second row, the first two lines of the sign that are highly prioritized, are fully reconstructed, while the third line, medium prioritized, is only partially reconstructed. These results are better viewed when zoomed in.

our priority-aware variant successfully reconstructs regions that would otherwise be poorly recovered, adapting fidelity according to the prioritization mask while keeping the same BPP. Additional details and results can be found in App. A.

Our second variant of Turbo-DDCM addresses the significant distortion variation across images at fixed target bitrates – see Fig. S19. We present a simple and efficient method to address this problem, substantially reducing this variation while targeting a given distortion level. See App. G for more details on this variant of the algorithm.

## 7 CONCLUSION AND DISCUSSION

This paper introduces Turbo-DDCM, an efficient and flexible zero-shot diffusion-based compression method. It achieves state-of-the-art runtime, reducing round-trip compression-decompression time by nearly an order of magnitude compared to the fastest prior zero-shot method, while maintaining competitive compression quality compared to state-of-the-art techniques. It also offers favorable properties, such as a constant bitrate across images for a given target bitrate. Moreover, we extend Turbo-DDCM with two variants: the first supports priority-aware compression for spatially-varying fidelity, and the second is a distortion-targeted mode that fixes PSNR instead of bitrate.

While our method significantly advances zero-shot compression speed, training-based methods can achieve similar reconstruction quality in a single forward pass. A one-step zero-shot method could provide significant additional speedup while preserving zero-shot flexibility. Moreover, some non-zero-shot methods demonstrate superior rate-distortion-perception trade-off compared to our method, suggesting potential room for improvement. Finally, establishing a comprehensive theory for DDCM-based compression remains an open problem that could guide future improvements.

ACKNOWLEDGMENTS

This research was partially supported by the Israel Science Foundation (ISF) under Grants 951/24, 409/24, and 2318/22, as well as by the Council for Higher Education – Planning and Budgeting Committee. G.O. gratefully acknowledges the Viterbi Fellowship from the Faculty of Electrical and Computer Engineering at the Technion.

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

APPENDIX

## A  EXPERIMENTAL CONFIGURATIONS AND ADDITIONAL EVALUATIONS

All distortion and perceptual quality metrics are computed using Torch Metrics, which is built on Torch Fidelity (Obukhov et al., 2020).

### A.1  CONFIGURATIONS

The following outlines the experimental configurations used for the evaluations in Section 5.

**Zero-Shot Methods.**  For all methods we use Stable Diffusion 2.1 Base[1], based on the official stabilityai/stable-diffusion-2-1-base checkpoint from Hugging Face, using float16 precision. The remaining hyperparameters are determined as follows:

- For Turbo-DDCM we use $T = 30$, $K = 16{,}384$, $C = 1$ and $M$ values in the range $[45, 300]$. To determine $N$, we initially assume $N = 0$ and calculate a preliminary value $\text{BPP}_0$ using eq. (15). Then we partition the interval $[0.01, 0.15]$ into 70 logarithmically spaced bins and set

$$N = \max(\min(70 - \text{bin}(\text{BPP}_0) - 1, T - 2), 0),  \tag{S1}$$

  where $\text{bin}(\text{BPP}_0)$ denotes the index of the logarithmic bin containing $\text{BPP}_0$. The scheduler we use is based on the optimized scheduler from Vonderfecht & Liu (2025), adapted to our setting by selecting evenly spaced indices from the timestep list for our $T$. See App. E for the rationale behind the chosen hyperparameters.

- For PSC we use the hyperparameters that the author described (Elata et al., 2024), setting the number of measurements to $12 \cdot 2^i$ for $i = 0, ..., 8$.

- For DDCM we use $T \in \{500, 1000\}$, $K \in \{64, 128, \ldots, 8192\}$, $M \in \{1, 2, ..., 6\}$ and $C \in \{2, 3\}$ which aligns with the spirit of the hyperparameter configurations described by the authors (Ohayon et al., 2025).

- For DiffC we use the schedulers, $D_{KL}$ values and reconstruction timesteps suggested by the authors (Vonderfecht & Liu, 2025).

**Non-Zero-Shot Methods.**  The hyperparameters for the non-zero-shot methods are as follows:

- For CRDR-R and CRDR-D we use the quality factors of $\{0, 1, 2, 3, 4\}$, where CRDR-D uses $\beta = 0$ and CRDR-R uses $\beta = 3.84$, as recommended by the authors (Iwai et al., 2024).

- For ILLM we use the MS-ILLM pre-trained models available in the Official GitHub repository. We use msillm_quality_X for X = 2,3 and msillm_quality_vloY for Y = 1,2.

- For BPG we use quality factors $q \in \{51, 50, 48, 46, 42, 40, 29, 36, 34, 32\}$.

- For PerCo (SD) we use the three publicly available Stabe Diffusion 2.1 fine-tuned checkpoints from the Official GitHub repository, with the default hyperparameters.

- For HiFiC we test the low quality regime, using the checkpoint available in the official GitHub repository.

- For StableCodec we use the checkpoints stablecodec_ftX for X = 2,4,8,16,32.

- For DiffEIC we test the 1_2_X pre-trained weights for X = 1,2,4,8,16.

**Turbo-DDCM Priority-Aware Variant Configuration.**  To achieve extremely low bitrates, we modify the Turbo-DDCM configuration above by setting $T = 20$. In addition, we use a linear scheduler.

---

[1]Recent models such as FLUX are trained using optimal transport–based flow matching (FM) (Lipman et al., 2023). To apply Turbo-DDCM, we translate between DDPM and OT flow model parameterizations, following an approach similar to Vonderfecht & Liu (2025). This extension is available in our code.

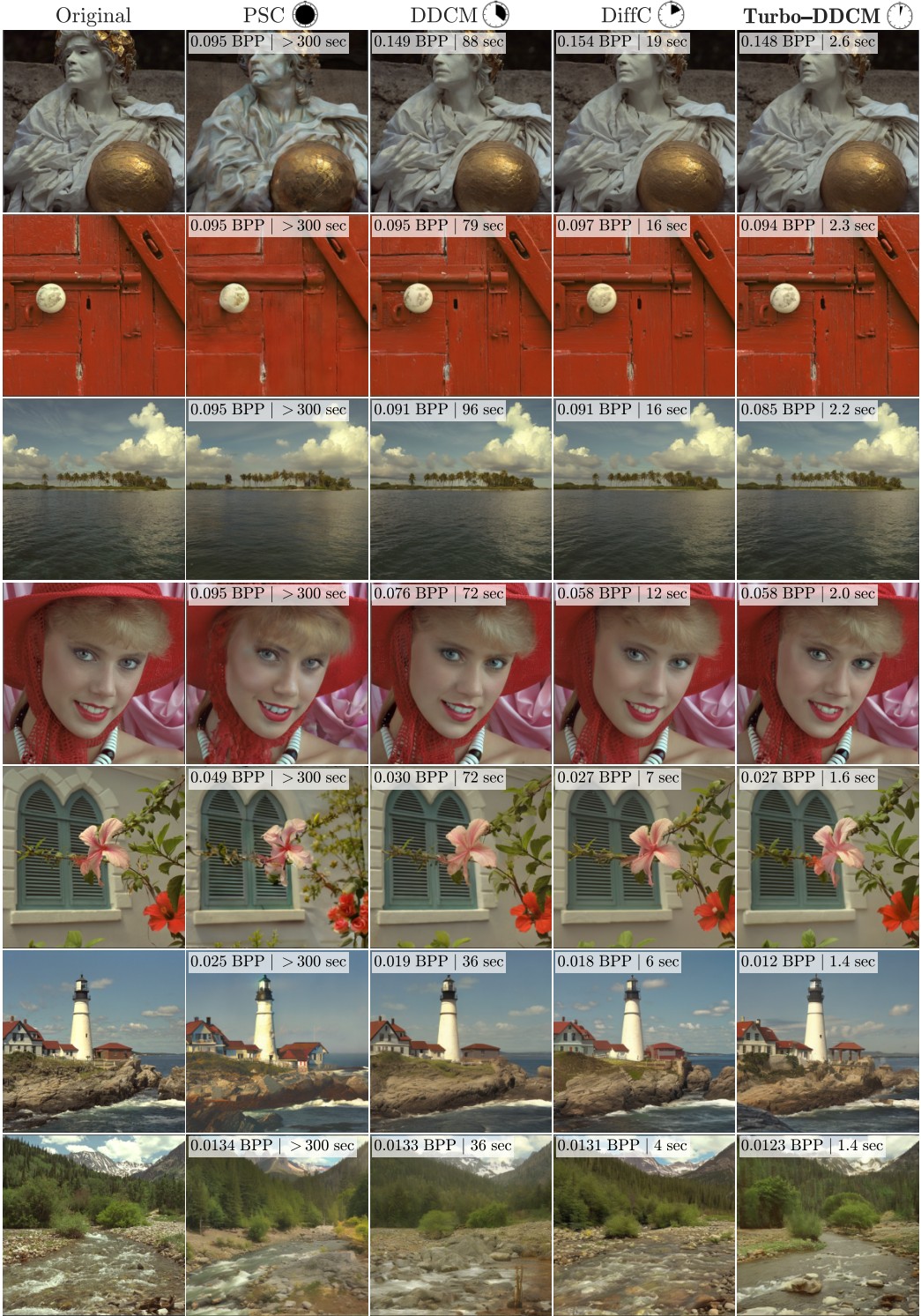

Figure S1: **Qualitative results compared to zero-shot methods:** The images are taken from the Kodak24 dataset, center-cropped to $512 \times 512$ pixels. Our compression method generates highly realistic reconstructions, matching or surpassing prior approaches in fidelity to the original images, while consistently offering significant runtime improvements.

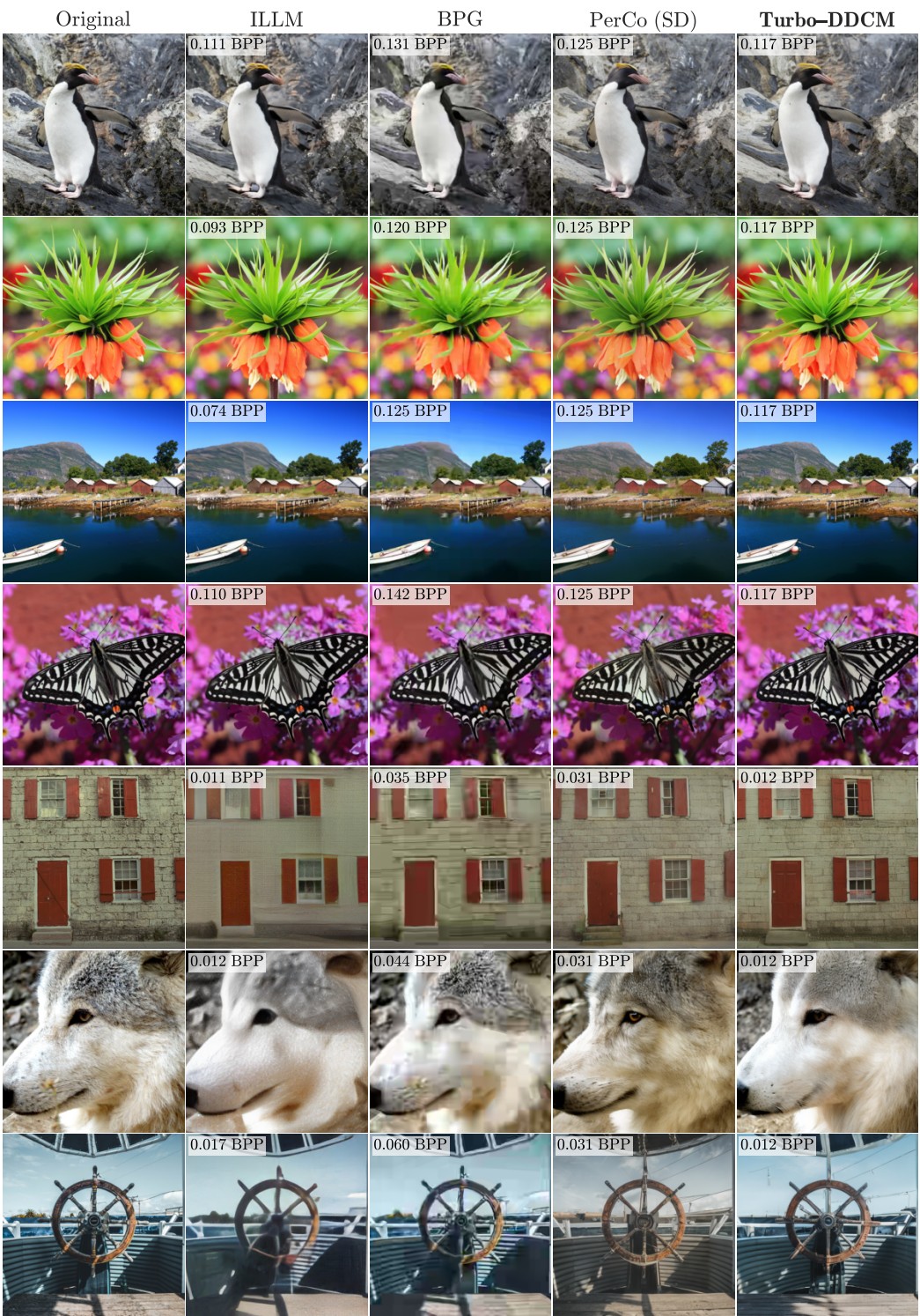

Figure S2: **Qualitative results compared to non-zero-shot methods:** The images are taken from the Kodak24 and DIV2K datasets, center-cropped to $512 \times 512$ pixels. Our compression method produces highly realistic reconstructions and preserves better fidelity to the original images compared to previous methods.

## A.2 Additional Efficiency Evaluation

Figure S3 decomposes the round-trip time results from Fig. 4 into compression and decompression components. The results demonstrate that our method's runtime advantage arises from balanced improvements in both processes, yielding faster performance for both the encoder and the decoder.

As discussed in Section 5, runtime measurements can be biased by implementation details. We therefore compare the number of neural function evaluations (NFEs) (Fig. S4), corresponding to denoiser activations, across the zero-shot methods. While NFEs are implementation-independent, they do not capture costs from other components, such as the RCC in DiffC, the MP process in DDCM and the thresholding-based process in Turbo-DDCM. The alignment between NFEs and runtime trends suggest that our efficiency is not attributable to implementation details.

## A.3 Additional Reconstruction Quality Evaluation

Figures S1 and S2 provide additional qualitative comparisons on the Kodak24 ($512 \times 512$) and DIV2K ($512 \times 512$) datasets, against both zero-shot and non-zero-shot methods. Figure S5 provides additional quantitative results. Tables 1, 2, 3 and 4 report the results presented in Fig. 4. Tables 5, 6 and 7 report BD-PSNR (Bjøntegaard, 2001).

## A.4 Additional Evaluations for the Priority-Aware Variant

Figure S6 illustrates how the priority-aware variant improves the fidelity of prioritized regions as the prioritization level increases. Figures S7 and S8 highlight the advantage of priority-aware compression in reconstructing complex image regions at extremely low bitrates. Depending on the prioritization level, our method successfully reconstructs these regions, while other methods fail to reconstruct them at all. In addition, our priority-aware compression method can also be applied to DDCM and DiffC, owing to the shared foundations of these approaches (see App. H). Figure S9 illustrates this generalization.

## A.5 GPU Memory Considerations

At first glance, Turbo-DDCM may appear to be a GPU-memory-bound algorithm. However, the only substantial memory overhead beyond the diffusion model itself comes from storing the codebooks. Using `float16` precision, Stable Diffusion 2.1-Base occupies approximately $2.5$ GB of GPU memory. Turbo-DDCM requires only a single codebook to be resident in memory at any given time. Each codebook contains 16k atoms, each of dimensionality 16k (SD 2.1-Base latent space dimensionality), stored in `float16`:

$$\text{codebook memory} = 16\text{k} \times 16\text{k} \times 2 \text{ bytes} = 2^{29} \text{ bytes} = 0.5 \text{ GB}.$$

Thus, the total expected peak memory for Turbo-DDCM is

$$2.5 \text{ GB (model)} + 0.5 \text{ GB (codebook)} \approx 3.0 \text{ GB}.$$

This matches our empirical measurements, which show a peak memory usage of approximately $3.05$ GB. In practice, Turbo-DDCM introduces only about a $20\%$ memory overhead relative to the diffusion model alone.

| method | bpp | PSNR | LPIPS | FID | Roundtrip Time (sec) |
|---|---|---|---|---|---|
| DDCM | 0.010 | 19.31 | 0.46 | 60.17 | 35.89 |
| | 0.013 | 20.46 | 0.36 | 48.84 | 35.74 |
| | 0.019 | 21.38 | 0.28 | 38.53 | 36.50 |
| | 0.025 | 21.92 | 0.23 | 33.53 | 35.92 |
| | 0.034 | 22.33 | 0.20 | 30.67 | 71.20 |
| | 0.046 | 22.84 | 0.17 | 26.87 | 73.37 |
| | 0.080 | 23.23 | 0.15 | 25.35 | 75.48 |
| | 0.114 | 23.48 | 0.14 | 25.21 | 74.27 |
| | 0.126 | 23.70 | 0.13 | 23.67 | 75.45 |
| | 0.149 | 24.04 | 0.12 | 22.59 | 87.95 |
| | 0.263 | 24.36 | 0.12 | 22.43 | 86.66 |
| DiffC | 0.007 | 18.66 | 0.41 | 40.49 | 3.57 |
| | 0.012 | 20.19 | 0.32 | 36.67 | 4.52 |
| | 0.019 | 21.26 | 0.25 | 32.98 | 5.58 |
| | 0.029 | 22.42 | 0.20 | 29.87 | 7.18 |
| | 0.050 | 23.73 | 0.15 | 26.69 | 10.19 |
| | 0.053 | 23.92 | 0.14 | 25.96 | 10.80 |
| | 0.057 | 24.08 | 0.13 | 26.39 | 11.23 |
| | 0.061 | 24.27 | 0.13 | 25.97 | 11.91 |
| | 0.066 | 24.44 | 0.12 | 25.46 | 12.43 |
| | 0.073 | 24.67 | 0.12 | 24.56 | 13.15 |
| | 0.081 | 24.89 | 0.11 | 24.00 | 14.38 |
| | 0.092 | 25.11 | 0.10 | 23.45 | 15.63 |
| | 0.109 | 25.42 | 0.09 | 22.69 | 17.09 |
| | 0.141 | 25.81 | 0.08 | 22.06 | 19.38 |
| PSC | 0.025 | 18.52 | 0.47 | 47.34 | - |
| | 0.048 | 20.08 | 0.37 | 40.70 | - |
| | 0.095 | 21.68 | 0.27 | 35.32 | - |
| Turbo-DDCM | 0.007 | 19.01 | 0.42 | 45.92 | 1.26 |
| | 0.012 | 20.16 | 0.33 | 38.92 | 1.36 |
| | 0.018 | 21.04 | 0.28 | 36.62 | 1.43 |
| | 0.031 | 22.33 | 0.21 | 30.83 | 1.63 |
| | 0.044 | 23.14 | 0.17 | 27.41 | 1.78 |
| | 0.058 | 23.82 | 0.14 | 25.75 | 1.97 |
| | 0.071 | 24.21 | 0.13 | 24.19 | 2.01 |
| | 0.085 | 24.58 | 0.12 | 23.15 | 2.19 |
| | 0.117 | 25.19 | 0.10 | 22.44 | 2.34 |
| | 0.148 | 25.55 | 0.09 | 22.07 | 2.53 |

Table 1: **Quantitative comparison on Kodak24$_{512}$ dataset between zero-shot methods.**

| method | bpp | PSNR | LPIPS | FID | Roundtrip Time (sec) |
|---|---|---|---|---|---|
| DDCM | 0.010 | 17.09 | 0.45 | 64.94 | 36.62 |
| | 0.013 | 18.01 | 0.37 | 52.43 | 35.96 |
| | 0.019 | 19.02 | 0.30 | 42.25 | 36.23 |
| | 0.025 | 19.64 | 0.25 | 36.97 | 36.86 |
| | 0.034 | 20.11 | 0.22 | 34.19 | 70.66 |
| | 0.046 | 20.71 | 0.19 | 31.61 | 73.59 |
| | 0.080 | 21.13 | 0.17 | 30.06 | 75.01 |
| | 0.114 | 21.41 | 0.16 | 29.54 | 77.24 |
| | 0.126 | 21.68 | 0.15 | 29.02 | 78.45 |
| | 0.149 | 22.10 | 0.13 | 28.26 | 87.17 |
| | 0.263 | 22.47 | 0.12 | 27.67 | 85.15 |
| DiffC | 0.008 | 16.68 | 0.39 | 43.84 | 3.52 |
| | 0.013 | 17.95 | 0.31 | 38.56 | 4.52 |
| | 0.020 | 19.07 | 0.26 | 35.86 | 5.57 |
| | 0.031 | 20.25 | 0.21 | 33.20 | 7.15 |
| | 0.053 | 21.76 | 0.15 | 29.58 | 10.21 |
| | 0.056 | 21.92 | 0.15 | 29.52 | 10.69 |
| | 0.060 | 22.12 | 0.14 | 28.73 | 11.22 |
| | 0.065 | 22.32 | 0.14 | 28.70 | 11.79 |
| | 0.070 | 22.54 | 0.13 | 28.24 | 12.39 |
| | 0.077 | 22.78 | 0.12 | 27.51 | 13.22 |
| | 0.086 | 23.05 | 0.12 | 27.23 | 14.24 |
| | 0.097 | 23.35 | 0.11 | 26.62 | 15.62 |
| | 0.115 | 23.71 | 0.10 | 25.92 | 17.12 |
| | 0.148 | 24.17 | 0.09 | 24.95 | 19.36 |
| PSC | 0.008 | 13.66 | 0.59 | 56.50 | - |
| | 0.013 | 14.91 | 0.53 | 51.51 | - |
| | 0.025 | 15.87 | 0.47 | 48.36 | - |
| | 0.048 | 17.47 | 0.39 | 44.27 | - |
| | 0.095 | 19.22 | 0.28 | 38.41 | - |
| Turbo-DDCM | 0.007 | 16.70 | 0.42 | 50.95 | 1.26 |
| | 0.012 | 17.83 | 0.34 | 44.41 | 1.34 |
| | 0.018 | 18.69 | 0.30 | 40.23 | 1.45 |
| | 0.031 | 20.10 | 0.23 | 36.22 | 1.65 |
| | 0.044 | 20.96 | 0.19 | 33.04 | 1.82 |
| | 0.065 | 21.96 | 0.15 | 29.54 | 2.03 |
| | 0.085 | 22.63 | 0.13 | 28.06 | 2.21 |
| | 0.117 | 23.33 | 0.11 | 26.68 | 2.42 |
| | 0.148 | 23.81 | 0.10 | 25.97 | 2.62 |
| | 0.194 | 24.11 | 0.09 | 24.98 | 2.66 |
| | 0.272 | 24.35 | 0.08 | 24.43 | 2.81 |

Table 2: **Quantitative comparison on DIV2K$_{512}$ dataset between zero-shot methods.**

| method | bpp | PSNR | LPIPS | FID | Roundtrip Time (sec) |
|---|---|---|---|---|---|
| BPG | 0.037 | 24.21 | 0.54 | 148.31 | 0.16 |
| | 0.044 | 24.64 | 0.51 | 141.58 | 0.17 |
| | 0.062 | 25.54 | 0.46 | 128.09 | 0.18 |
| | 0.088 | 26.49 | 0.41 | 114.26 | 0.19 |
| | 0.121 | 27.38 | 0.36 | 104.29 | 0.21 |
| | 0.167 | 28.35 | 0.31 | 92.42 | 0.22 |
| | 0.227 | 29.36 | 0.26 | 82.18 | 0.24 |
| CRDR-D | 0.114 | 27.99 | 0.26 | 94.50 | 3.88 |
| | 0.210 | 30.07 | 0.17 | 76.04 | 3.89 |
| CRDR-R | 0.114 | 27.31 | 0.09 | 35.69 | 3.89 |
| | 0.210 | 29.37 | 0.06 | 28.40 | 3.89 |
| DiffEIC | 0.023 | 18.83 | 0.34 | 41.68 | 5.25 |
| | 0.043 | 20.27 | 0.25 | 35.61 | 5.25 |
| | 0.067 | 22.01 | 0.18 | 30.52 | 5.26 |
| | 0.098 | 23.63 | 0.13 | 26.54 | 5.26 |
| | 0.130 | 23.40 | 0.11 | 24.11 | 5.27 |
| HiFiC | 0.191 | 27.38 | 0.07 | 31.71 | 19.99 |
| ILLM | 0.009 | 20.27 | 0.50 | 128.40 | 0.32 |
| | 0.014 | 21.43 | 0.40 | 93.67 | 0.32 |
| | 0.100 | 25.68 | 0.11 | 36.08 | 0.32 |
| | 0.182 | 27.30 | 0.07 | 28.56 | 0.32 |
| PerCo (SD) | 0.031 | 19.02 | 0.31 | 37.02 | 2.68 |
| | 0.125 | 22.32 | 0.14 | 26.42 | 2.76 |
| StableCodec | 0.008 | 19.94 | 0.32 | 43.36 | 0.37 |
| | 0.013 | 20.98 | 0.25 | 37.91 | 0.37 |
| | 0.020 | 21.71 | 0.21 | 34.93 | 0.37 |
| | 0.029 | 22.34 | 0.18 | 32.91 | 0.37 |
| | 0.038 | 22.69 | 0.17 | 31.94 | 0.37 |
| Turbo-DDCM | 0.007 | 19.01 | 0.42 | 45.92 | 1.26 |
| | 0.012 | 20.16 | 0.33 | 38.92 | 1.36 |
| | 0.018 | 21.04 | 0.28 | 36.62 | 1.43 |
| | 0.031 | 22.33 | 0.21 | 30.83 | 1.63 |
| | 0.044 | 23.14 | 0.17 | 27.41 | 1.78 |
| | 0.058 | 23.82 | 0.14 | 25.75 | 1.97 |
| | 0.071 | 24.21 | 0.13 | 24.19 | 2.01 |
| | 0.085 | 24.58 | 0.12 | 23.15 | 2.19 |
| | 0.117 | 25.19 | 0.10 | 22.44 | 2.34 |
| | 0.148 | 25.55 | 0.09 | 22.07 | 2.53 |

Table 3: **Quantitative comparison on Kodak24$_{512}$ dataset between non-zero-shot methods.**

| method | bpp | PSNR | LPIPS | FID | Roundtrip Time (sec) |
|---|---|---|---|---|---|
| BPG | 0.058 | 22.79 | 0.49 | 152.62 | 0.19 |
| | 0.068 | 23.22 | 0.46 | 146.22 | 0.19 |
| | 0.096 | 24.12 | 0.41 | 134.03 | 0.20 |
| | 0.134 | 25.09 | 0.35 | 122.90 | 0.22 |
| | 0.181 | 26.03 | 0.29 | 110.13 | 0.23 |
| | 0.245 | 27.01 | 0.24 | 98.93 | 0.25 |
| CRDR-D | 0.153 | 26.62 | 0.20 | 89.78 | 3.86 |
| | 0.274 | 28.76 | 0.12 | 69.48 | 3.85 |
| CRDR-R | 0.153 | 26.11 | 0.08 | 39.30 | 3.84 |
| | 0.274 | 28.27 | 0.05 | 31.43 | 3.85 |
| DiffEIC | 0.023 | 16.75 | 0.33 | 42.77 | 5.21 |
| | 0.043 | 18.43 | 0.24 | 38.23 | 5.21 |
| | 0.067 | 20.26 | 0.18 | 33.03 | 5.23 |
| | 0.098 | 21.94 | 0.13 | 28.88 | 5.21 |
| | 0.130 | 21.84 | 0.11 | 27.68 | 5.19 |
| HiFiC | 0.226 | 26.14 | 0.07 | 39.83 | 20.26 |
| ILLM | 0.009 | 18.56 | 0.49 | 119.83 | 0.30 |
| | 0.014 | 19.74 | 0.38 | 96.94 | 0.30 |
| | 0.100 | 24.58 | 0.11 | 42.59 | 0.30 |
| | 0.182 | 26.21 | 0.07 | 34.10 | 0.30 |
| PerCo (SD) | 0.031 | 16.52 | 0.33 | 40.95 | 2.69 |
| | 0.125 | 20.48 | 0.16 | 29.52 | 2.71 |
| StableCodec | 0.008 | 17.95 | 0.32 | 45.98 | 0.35 |
| | 0.013 | 18.97 | 0.26 | 39.85 | 0.35 |
| | 0.020 | 19.70 | 0.22 | 36.72 | 0.35 |
| | 0.029 | 20.33 | 0.20 | 35.05 | 0.35 |
| | 0.038 | 20.72 | 0.18 | 33.95 | 0.35 |
| Turbo-DDCM | 0.007 | 16.70 | 0.42 | 50.95 | 1.26 |
| | 0.012 | 17.83 | 0.34 | 44.41 | 1.34 |
| | 0.018 | 18.69 | 0.30 | 40.23 | 1.45 |
| | 0.031 | 20.10 | 0.23 | 36.22 | 1.65 |
| | 0.044 | 20.96 | 0.19 | 33.04 | 1.82 |
| | 0.065 | 21.96 | 0.15 | 29.54 | 2.03 |
| | 0.085 | 22.63 | 0.13 | 28.06 | 2.21 |
| | 0.117 | 23.33 | 0.11 | 26.68 | 2.42 |
| | 0.148 | 23.81 | 0.10 | 25.97 | 2.62 |

Table 4: **Quantitative comparison on DIV2K$_{512}$ dataset between non-zero-shot methods.**

| A
B | DDCM | DiffC | PSC | Turbo-DDCM |
|---|---|---|---|---|
| DDCM | 0.00 | -0.98 | 2.69 | -0.76 |
| DiffC | 0.98 | 0.00 | 3.61 | 0.07 |
| PSC | -2.69 | -3.61 | 0.00 | -3.35 |
| Turbo-DDCM | 0.76 | -0.07 | 3.35 | 0.00 |

| A
B | BPG | CRDR-D | CRDR-R | DiffEIC | HiFiC | ILLM | PerCo (SD) | Stable-Codec | Turbo-DDCM |
|---|---|---|---|---|---|---|---|---|---|
| BPG | 0.00 | -0.99 | -0.22 | 3.75 | 1.67 | 1.02 | 4.95 | 1.58 | 1.76 |
| CRDR-D | 0.99 | 0.00 | 0.76 | 4.66 | 2.70 | 2.12 | 5.93 | 3.51 | 3.06 |
| CRDR-R | 0.22 | -0.76 | 0.00 | 3.98 | 1.96 | 1.43 | 5.25 | 3.03 | 2.37 |
| DiffEIC | -3.75 | -4.66 | -3.98 | 0.00 | -4.64 | -3.15 | 0.91 | -3.30 | -2.36 |
| HiFiC | -1.67 | -2.70 | -1.96 | 4.64 | 0.00 | -0.03 | 4.54 | 8.66 | 1.56 |
| ILLM | -1.02 | -2.12 | -1.43 | 3.15 | 0.03 | 0.00 | 4.05 | 0.52 | 0.86 |
| PerCo (SD) | -4.95 | -5.93 | -5.25 | -0.91 | -4.54 | -4.05 | 0.00 | -3.56 | -3.16 |
| StableCodec | -1.58 | -3.51 | -3.03 | 3.30 | -8.66 | -0.52 | 3.56 | 0.00 | 0.46 |
| Turbo-DDCM | -1.76 | -3.06 | -2.37 | 2.36 | -1.56 | -0.86 | 3.16 | -0.46 | 0.00 |

Table 5: **BD-PSNR evaluation on Kodak24$_{512}$ dataset**

| A
B | DDCM | DiffC | PSC | Turbo-DDCM |
|---|---|---|---|---|
| DDCM | 0.00 | -0.85 | 3.24 | -0.84 |
| DiffC | 0.85 | 0.00 | 3.41 | 0.27 |
| PSC | -3.24 | -3.41 | 0.00 | -3.53 |
| Turbo-DDCM | 0.84 | -0.27 | 3.53 | 0.00 |

| A
B | BPG | CRDR-D | CRDR-R | DiffEIC | HiFiC | ILLM | PerCo (SD) | Stable-Codec | Turbo-DDCM |
|---|---|---|---|---|---|---|---|---|---|
| BPG | 0.00 | -1.41 | -0.88 | 2.60 | NaN | -0.33 | 3.81 | 1.33 | 1.25 |
| CRDR-D | 1.41 | 0.00 | 0.52 | 4.62 | NaN | 0.96 | 5.79 | 3.46 | 2.72 |
| CRDR-R | 0.88 | -0.52 | 0.00 | 4.12 | NaN | 0.46 | 5.28 | 2.97 | 2.22 |
| DiffEIC | -2.60 | -4.62 | -4.12 | 0.00 | -13.98 | -3.67 | 0.85 | -3.26 | -2.07 |
| HiFiC | NaN | NaN | NaN | 13.98 | NaN | 2.71 | 15.00 | 44.03 | 8.28 |
| ILLM | 0.33 | -0.96 | -0.46 | 3.67 | -2.71 | 0.00 | 4.34 | 0.92 | 1.62 |
| PerCo (SD) | -3.81 | -5.79 | -5.28 | -0.85 | -15.00 | -4.34 | 0.00 | -3.57 | -2.76 |
| StableCodec | -1.33 | -3.46 | -2.97 | 3.26 | -44.03 | -0.92 | 3.57 | 0.00 | 0.77 |
| Turbo-DDCM | -1.25 | -2.72 | -2.22 | 2.07 | -8.28 | -1.62 | 2.76 | -0.77 | 0.00 |

Table 6: **BD-PSNR evaluation on DIV2K$_{512}$ dataset**

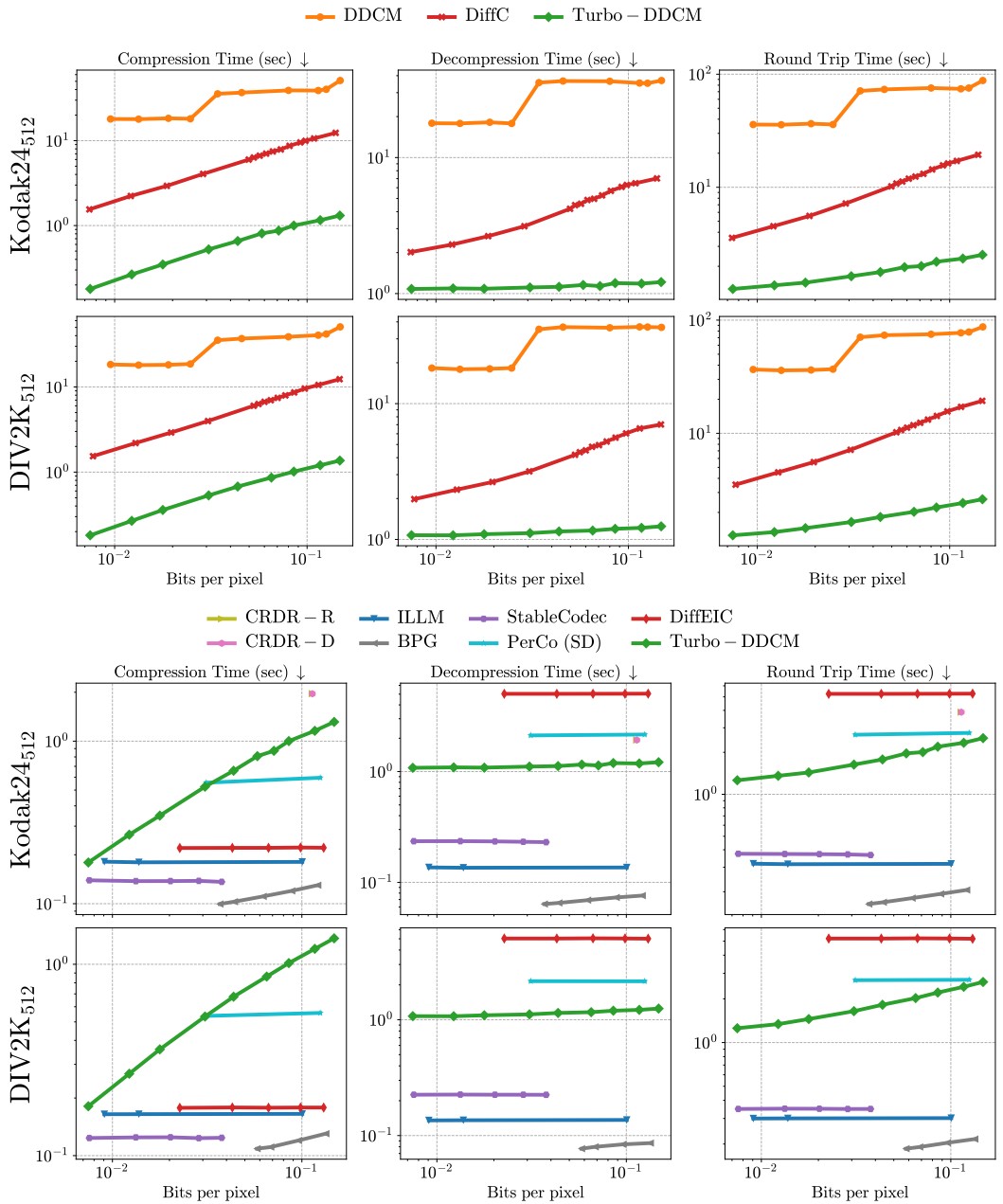

Figure S3: **Compression & Decompression runtime comparison.**

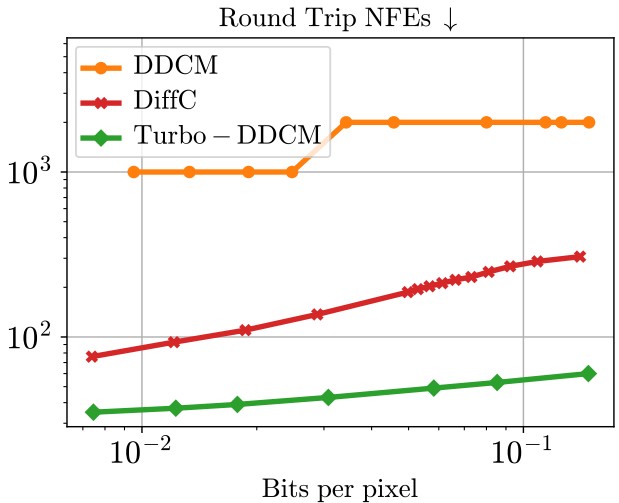

Figure S4: **Round-trip compression-decompression NFEs comparison:** Our method maintains its advantage over others, with margins similar to those observed in runtime.

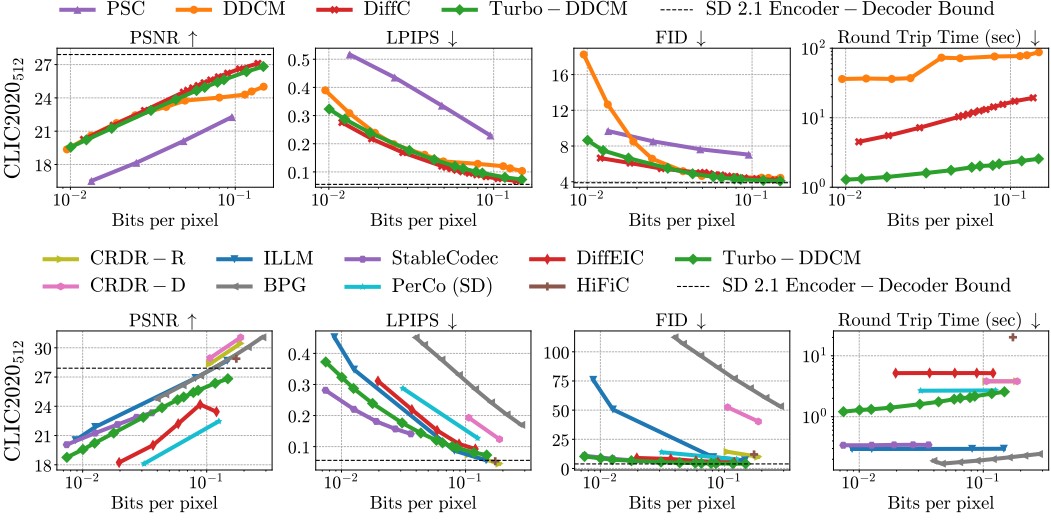

Figure S5: **Additional Quantitative Results.**

| A
B | DDCM | DiffC | PSC | Turbo-DDCM |
|---|---|---|---|---|
| DDCM | 0.00 | -0.84 | 3.71 | -0.84 |
| DiffC | 0.84 | 0.00 | 4.07 | 0.14 |
| PSC | -3.71 | -4.07 | 0.00 | -3.88 |
| Turbo-DDCM | 0.84 | -0.14 | 3.88 | 0.00 |

| A
B | BPG | CRDR-D | CRDR-R | DiffEIC | HiFiC | ILLM | PerCo (SD) | Stable-Codec | Turbo-DDCM |
|---|---|---|---|---|---|---|---|---|---|
| BPG | 0.00 | -1.53 | -0.87 | 3.60 | NaN | -0.08 | 5.08 | 1.27 | 1.40 |
| CRDR-D | 1.53 | 0.00 | 0.66 | 5.42 | NaN | 1.36 | 6.83 | 3.37 | 3.04 |
| CRDR-R | 0.87 | -0.66 | 0.00 | 4.83 | NaN | 0.76 | 6.24 | 3.02 | 2.44 |
| DiffEIC | -3.60 | -5.42 | -4.83 | 0.00 | -11.59 | -4.35 | 1.35 | -4.14 | -2.84 |
| HiFiC | NaN | NaN | NaN | 11.59 | NaN | 2.10 | 10.78 | 33.29 | 3.93 |
| ILLM | 0.08 | -1.36 | -0.76 | 4.35 | -2.10 | 0.00 | 5.32 | 0.91 | 1.55 |
| PerCo (SD) | -5.08 | -6.83 | -6.24 | -1.35 | -10.78 | -5.32 | 0.00 | -4.68 | -3.86 |
| StableCodec | -1.27 | -3.37 | -3.02 | 4.14 | -33.29 | -0.91 | 4.68 | 0.00 | 0.76 |
| Turbo-DDCM | -1.40 | -3.04 | -2.44 | 2.84 | -3.93 | -1.55 | 3.86 | -0.76 | 0.00 |

Table 7: **BD-PSNR evaluation on CLIC2020$_{512}$**

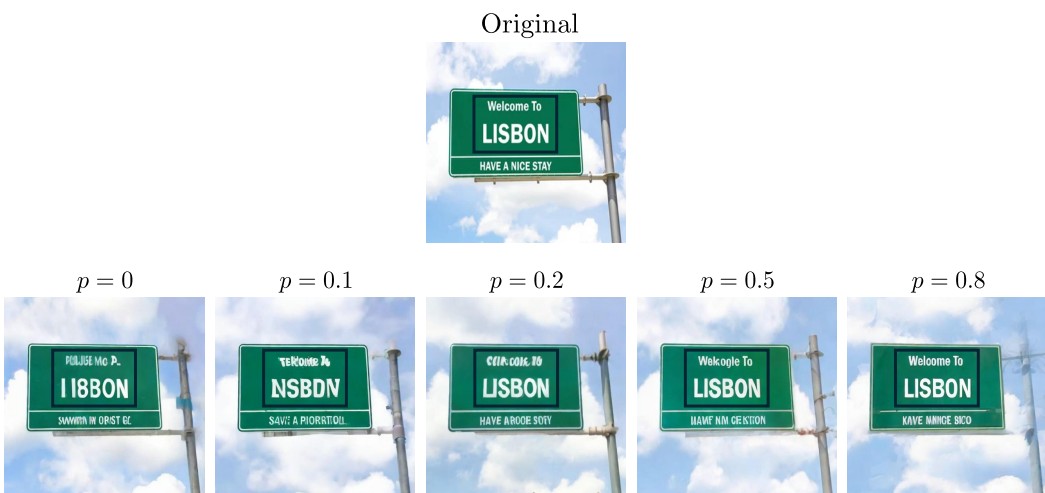

Figure S6: **Priority-Aware results:** $p$ stands for the prioritization level, such that the values in $\boldsymbol{w}$ (Section 6) are 1 for the deprioritized pixels and $1 + p$ for the prioritized ones. Reconstruction of the prioritized regions shows better fidelity to the original image as the prioritization level increases.

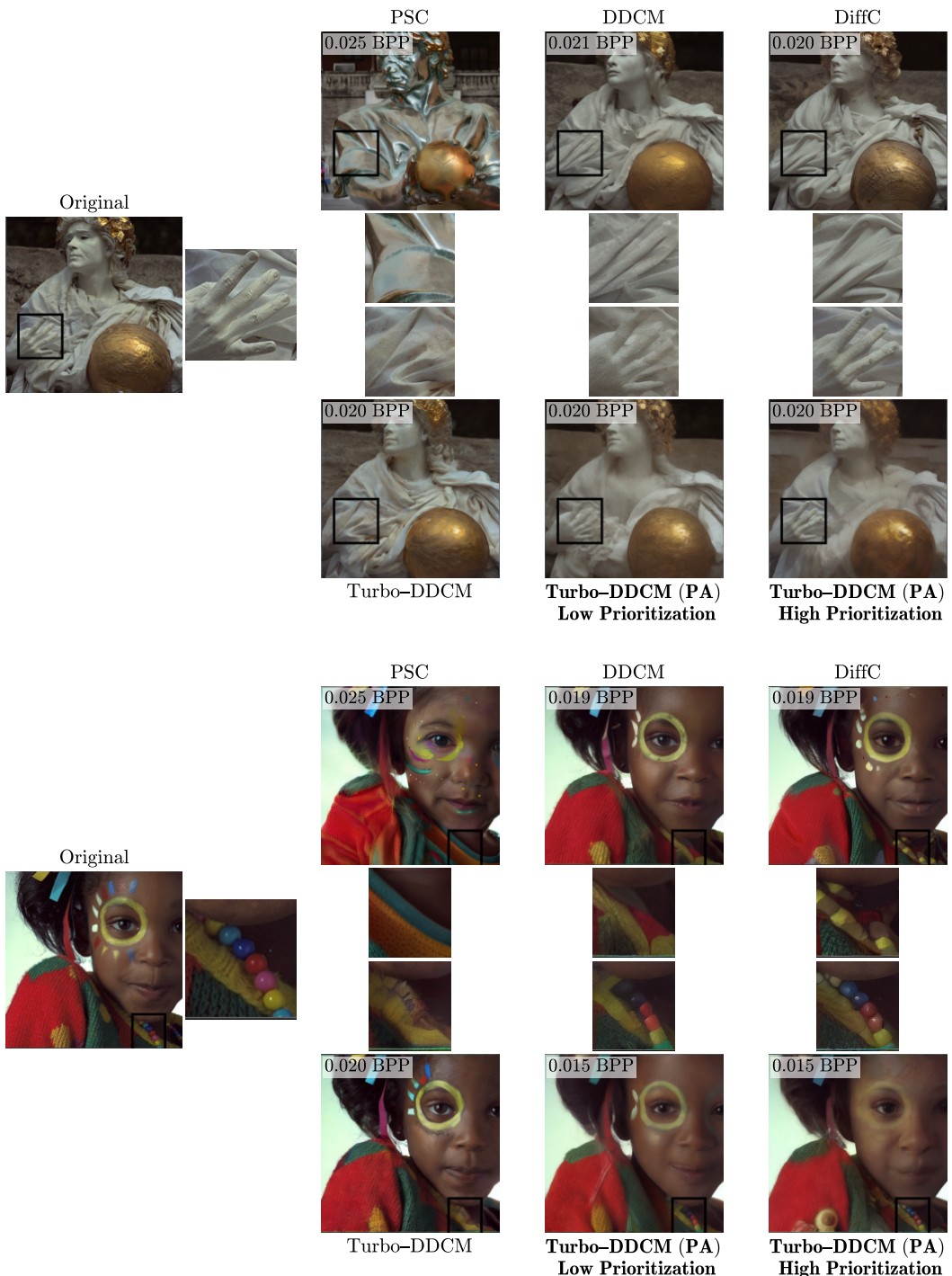

Figure S7: **Priority-Aware results:** With our priority-aware (PA) variant, specific objects in the image can be reconstructed at extremely low BPPs, with fidelity depending on the prioritization level. These objects are not reconstructed by other methods at the same BPP.

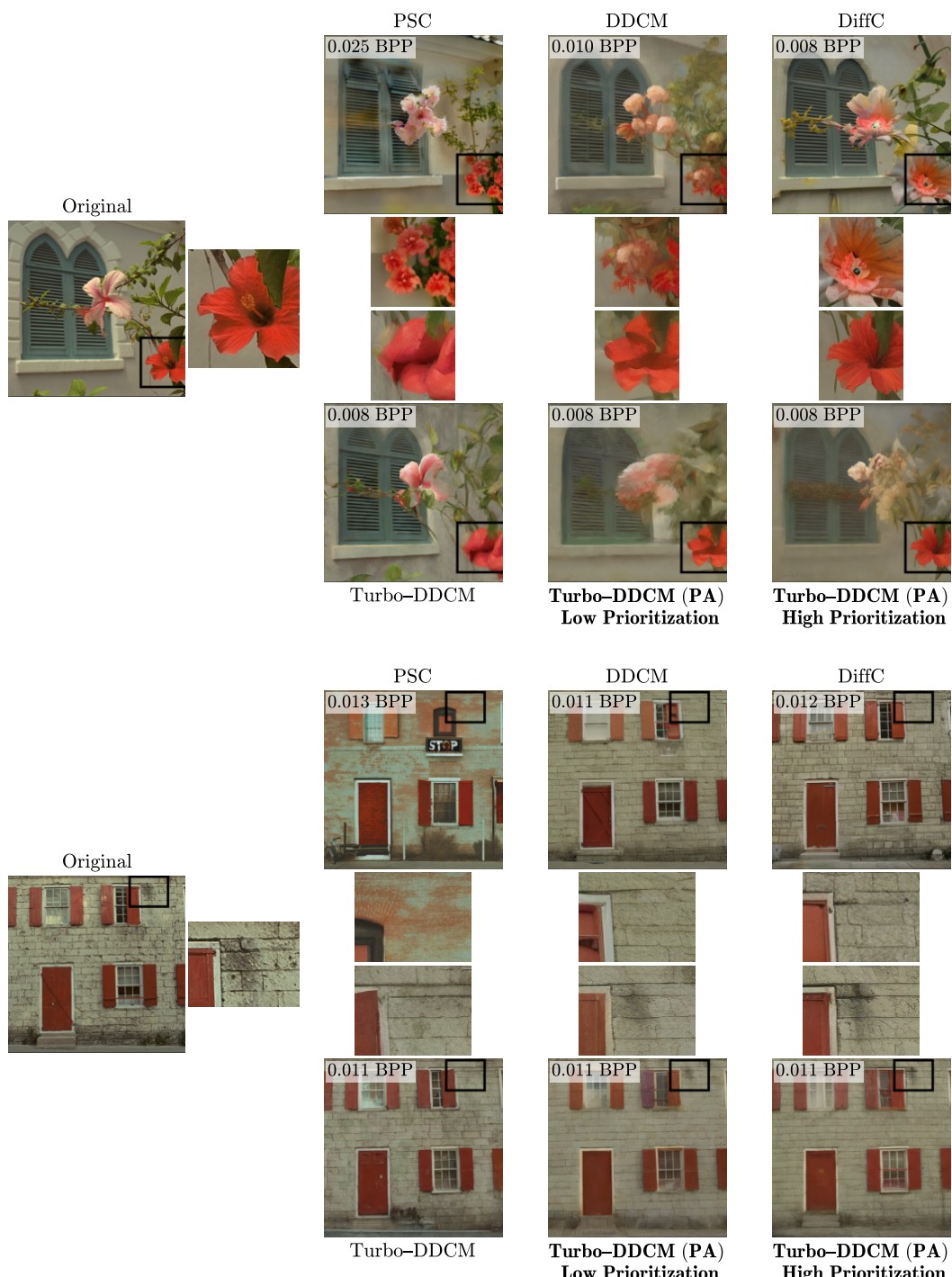

Figure S8: **Priority-Aware results:** With our priority-aware (PA) variant, specific objects in the image can be reconstructed at extremely low BPPs, with fidelity depending on the prioritization level. These objects are barely reconstructed, if at all, by other methods at the same BPP.

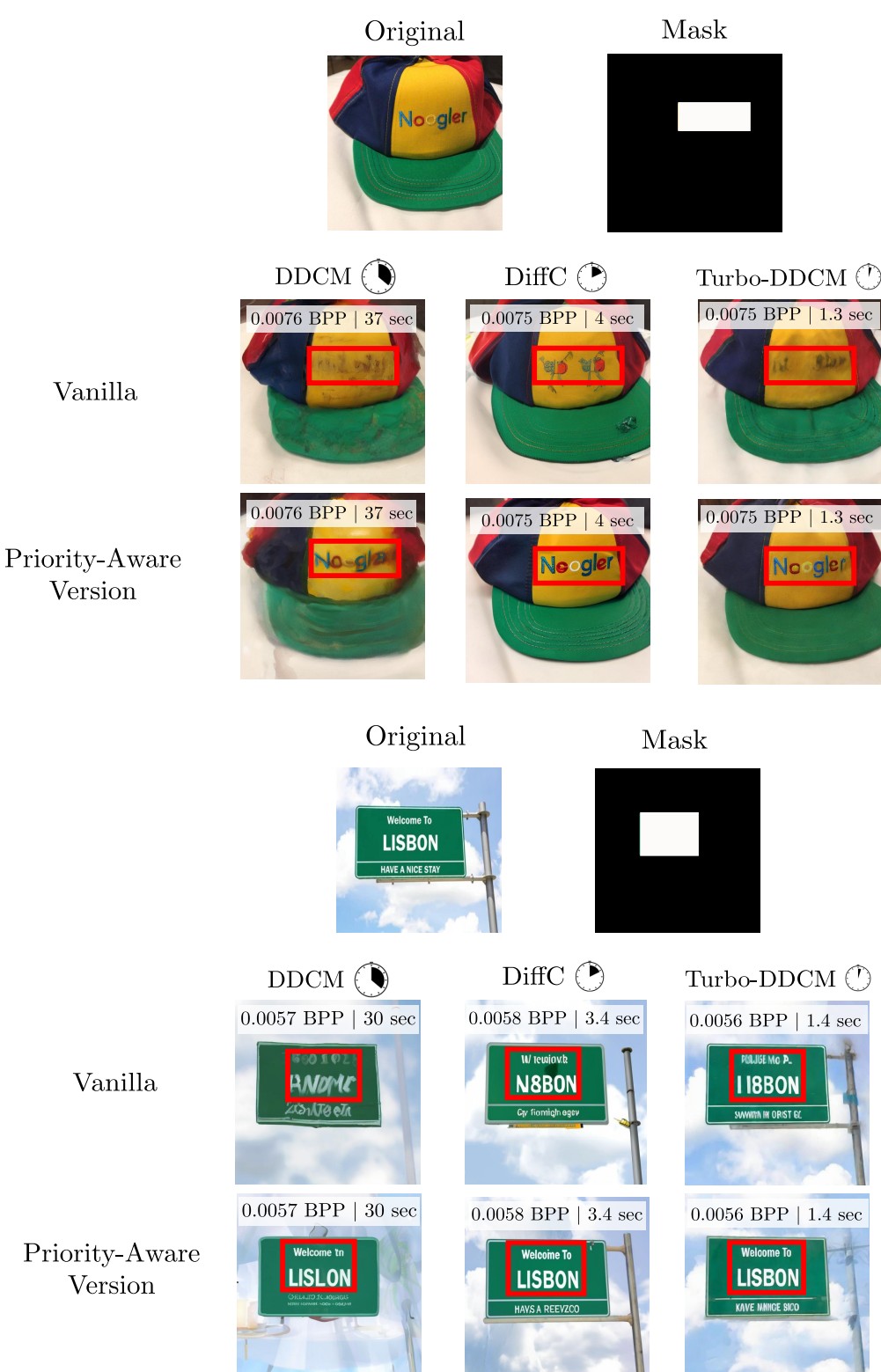

Figure S9: **Priority-Aware results in DDCM and DiffC:** Our method for priority-aware can be extended to DDCM and DiffC. Turbo-DDCM preserves its superiority in runtime.

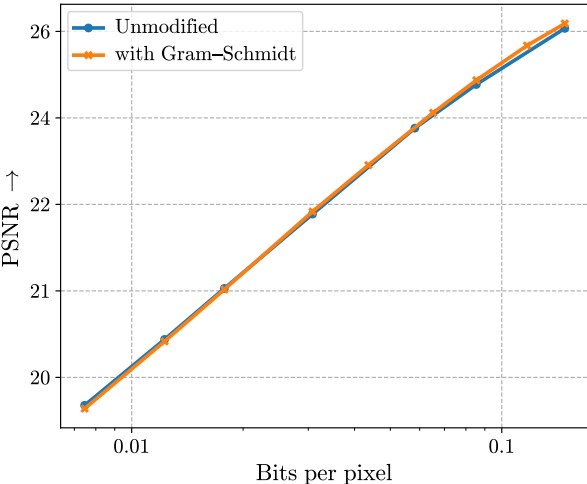

Figure S10: **Effect of the orthogonality assumption:** The codebook size used is $16k \times 16k$. Applying Gram-Schmidt orthogonalization to the codebook does not improve rate-distortion compared to the unmodified version, which does not apply Gram-Schmidt.

## B    CODEBOOKS NEAR-ORTHOGONALITY

As described in Section 4, we assume that the codebook in each diffusion step $C_t$ is orthogonal to build on the efficient and closed-form thresholding algorithm. However, since the codebook consists of i.i.d. Gaussian noise vectors, it is not necessarily orthogonal. Thus, in this short appendix, we evaluate how much this approximation degrades the quality of the solution. We do so by applying Gram-Schmidt (GS) orthogonalization to all the codebooks and comparing the results to the baseline, which does not use GS. Notably, if GS is performed in the encoder, it should be done in the decoder as well. On the one hand, GS requires substantial computation, since we may use large codebooks with more than $K = 16k$ atoms, this results in codebooks larger than $16k \times 16k$ in the SD-2.1-Base latent space. On the other hand, GS makes the orthogonality assumption valid and might yield better solutions that could enable better compression capabilities.

Figure S10 shows that for our chosen codebook size of $K = 16{,}384$ atoms, the rate-distortion without GS is almost identical to the rate-distortion with GS.

## C THRESHOLDING-BASED STRATEGY CORRECTNESS

As explained in Section 4, Turbo-DDCM employs a fundamentally different approach to combining codebook atoms compared to DDCM. Our method builds on the thresholding algorithm (Elad, 2010), which solves sparse least squares, with modifications tailored to our problem, such as the quantization constraint. To present our considerations clearly, we derive the solution from scratch. We also compare our approach to DDCM MP.

### C.1 PROBLEM SETUP

**Inputs.** For diffusion timestep $t$:

- Original image: $\mathbf{x}_0$
- Prediction of the original image: $\hat{\mathbf{x}}_{0|t}$
- Codebook of i.i.d. Gaussian atoms: $\boldsymbol{C}_t \in \mathbb{R}^{d \times K}$, where $d$ is the dimensionality and $K$ is the codebook size
- Sparsity parameter: $M$
- Quantization ordered set: $\mathcal{V} = [q_0, ..., q_{2^C}]$ (communication cost is $\log_2 |\mathcal{V}| = C$ bits).

We define the residual at timestep $t$ as

$$\boldsymbol{r}_t := \mathbf{x}_0 - \hat{\mathbf{x}}_{0|t}. \tag{S2}$$

**Objective:** Find a linear combination of codebook atoms that best approximates the residual, subject to a sparsity constraint and a quantization constraint. Formally, the optimization problem is

$$\mathbf{s}_t^* = \underset{\mathbf{s}_t \in \mathbb{R}^K}{\arg\min} \|\boldsymbol{C}_t \mathbf{s}_t - \mathbf{r}_t\|_2^2 \quad \text{s.t.} \quad \|\mathbf{s}_t\|_0 = M, \ \ \forall i \, (\mathbf{s}_t)_i \in \mathcal{V} \cup \{0\}. \tag{S3}$$

**Assumption:** $\boldsymbol{C}_t$ is nearly-orthogonal. See App. B for the impact of this assumption.

### C.2 APPROXIMATED CLOSED-FORM DERIVATION

Let $\text{proj}_t$ be the orthogonal projection of $\boldsymbol{r}_t$ onto the column space of $\boldsymbol{C}_t$. By the Pythagorean theorem,

$$\|\boldsymbol{C}_t \mathbf{s}_t - \boldsymbol{r}_t\|_2^2 = \|\boldsymbol{C}_t \mathbf{s}_t - \text{proj}_t\|_2^2 + \|\text{proj}_t - \boldsymbol{r}_t\|_2^2. \tag{S4}$$

Since $\|\text{proj}_t - \mathbf{r}_t\|_2^2$ is independent of $\mathbf{s}_t$, the objective reduces to

$$\underset{\mathbf{s}_t}{\arg\min} \|\boldsymbol{C}_t \mathbf{s}_t - \text{proj}_t\|_2^2, \tag{S5}$$

subject to the same constraints.

Because $\text{proj}_t$ lies in the column space of $\boldsymbol{C}_t$, there exists $\mathbf{s}_{proj}$ such that $\text{proj}_t = \boldsymbol{C}_t \mathbf{s}_{proj}$, yielding

$$\underset{\mathbf{s}_t}{\arg\min} \|\boldsymbol{C}_t (\mathbf{s}_t - \mathbf{s}_{\text{proj}})\|_2^2. \tag{S6}$$

Under the near-orthogonality assumption

$$\mathbf{s}_{\text{proj},i} \approx \frac{\langle \boldsymbol{z}_t^{(i)}, \boldsymbol{r}_t \rangle}{\|\boldsymbol{z}_t^{(i)}\|_2^2}, \tag{S7}$$

where $\boldsymbol{z}_t^{(i)}$ is the $i$-th column of $\boldsymbol{C}_t$. Since the atoms are high-dimensional i.i.d. Gaussian vectors, their norms are approximately equal. Hence, the optimization is approximated by

$$\underset{\mathbf{s}_t}{\arg\min} \left\| \sum_{i=1}^{K} (\mathbf{s}_{t,i} - \langle \boldsymbol{z}_t^{(i)}, \boldsymbol{r}_t \rangle) \boldsymbol{z}_i \right\|_2^2. \tag{S8}$$

Using the near-orthogonality and the Pythagorean theorem the problem reduces to

$$\underset{\mathbf{s}_t}{\arg\min} \sum_{i=1}^{K} (\mathbf{s}_{t,i} - \langle \boldsymbol{z}_t^{(i)}, \boldsymbol{r}_t \rangle)^2 \|\boldsymbol{z}_i\|_2^2 \approx \underset{\mathbf{s}_t}{\arg\min} \sum_{i=1}^{K} (\mathbf{s}_{t,i} - \langle \boldsymbol{z}_t^{(i)}, \boldsymbol{r}_t \rangle)^2. \tag{S9}$$

**Sparsity only constraint.** If we ignore quantization constraint and enforce only the sparsity constraint, then for any support $S \subset \{1, \ldots, K\}$ with $|S| = M$ the optimal coefficients are

$$\mathbf{s}_{t,i} = \begin{cases} \alpha_i := \langle \boldsymbol{z}_t^{(i)}, \boldsymbol{r}_t \rangle, & i \in S, \\ 0, & i \notin S. \end{cases} \tag{S10}$$

The corresponding error is

$$E(S) = \sum_{i \notin S} \alpha_i^2. \tag{S11}$$

Thus, the best support is obtained by selecting the $M$ indices with the largest values of $\alpha_i^2$.

**Adding the quantization constraint.** Now coefficients must lie in the finite set $\mathcal{V}$. For a fixed index $i$, the best quantized value is the nearest element of $\mathcal{V}$ to $\alpha_i$. Thus, we define

$$q_i^\star := \arg\min_{q \in \mathcal{V}} |q - \alpha_i|. \tag{S12}$$

If we select index $i$ and quantize to $q_i^\star$, the error contribution of index $i$ becomes

$$E_i^{\text{quant}} = (q_i^\star - \alpha_i)^2. \tag{S13}$$

Compared to leaving index $i$ out (which yields error of $\alpha_i^2$), the net reduction in error by selecting-and-quantizing $i$ is

$$\Delta_i = \alpha_i^2 - (q_i^\star - \alpha_i)^2. \tag{S14}$$

As a result, the best support in that case is obtained by selecting the $M$ indices with the largest values of $(\alpha_i^2 - (q_i^\star - \alpha_i)^2)$.

**Turbo-DDCM Characteristic Quantization Levels.** In Turbo-DDCM we use the quantization set $\mathcal{V} = [-1, 1]$, corresponding to $C = 1$ (see App. E for the rationale). Thus, for each atom, the nearest quantized value is $q_i^\star = \text{sign}(\alpha_i)$, yielding a net error reduction

$$\Delta_i = 2|\alpha_i| - 1 = 2|\langle \boldsymbol{z}_t^{(i)}, \boldsymbol{r}_t \rangle| - 1. \tag{S15}$$

Thus, the support can be chosen simply by selecting the $M$ indices with the largest $|\langle \boldsymbol{z}_t^{(i)}, \boldsymbol{r}_t \rangle|$ values with correspondence quantized coefficients of $\text{sign}(\langle \boldsymbol{z}_t^{(i)}, \boldsymbol{r}_t \rangle)$.

### C.3 Approximation Quality and Hyperparameter: Turbo-DDCM vs. DDCM MP

Both DDCM MP and our thresholding-based strategy aim to combine codebook atoms to improve the approximation of the residual vector. Although their approaches fundamentally differ, their underlying goal is the same. This becomes evident when evaluating the final constructed noise $\mathbf{z}^*$ from each method within their respective optimization problems for $M = 1$, assuming $\mathbf{z}^*$ is the selected noise. In DDCM, the objective reduces to maximizing the inner product between the noise and the residual vector, while in Turbo-DDCM it reduces to minimizing the $L_2$ norm of their difference (eq. (6) and eq. (9)). Importantly, $\|\mathbf{z}^*\|_2$ is approximately constant across both methods, due to normalization to unit standard deviation and the approximate zero mean of $\mathbf{z}^*$. Under this condition, the objectives are closely related:

$$\arg\min_{\mathbf{z}^*} \|\mathbf{z}^* - \boldsymbol{r}\|_2^2 = \arg\min_{\mathbf{z}^*} \left( \|\mathbf{z}^*\|_2^2 + \|\boldsymbol{r}\|_2^2 - 2\langle \mathbf{z}^*, \boldsymbol{r} \rangle \right) \tag{S16}$$

$$\approx \arg\max_{\mathbf{z}^*} \langle \mathbf{z}^*, \boldsymbol{r} \rangle. \tag{S17}$$

Moreover, since $\langle \mathbf{z}^*, \boldsymbol{r} \rangle = \|\mathbf{z}^*\|_2 \|\boldsymbol{r}\|_2 \cos\theta$, with $\theta \in [0, \pi]$ denoting the unsigned angle between the two vectors, we obtain

$$\arg\max_{\mathbf{z}^*} \langle \mathbf{z}^*, \boldsymbol{r} \rangle = \arg\max_{\mathbf{z}^*} \|\mathbf{z}^*\|_2 \|\boldsymbol{r}\|_2 \cos\theta \tag{S18}$$

$$\approx \arg\min_{\mathbf{z}^*} \theta \qquad (\|\mathbf{z}^*\|_2 \|\boldsymbol{r}\|_2 \approx \text{const} > 0). \tag{S19}$$

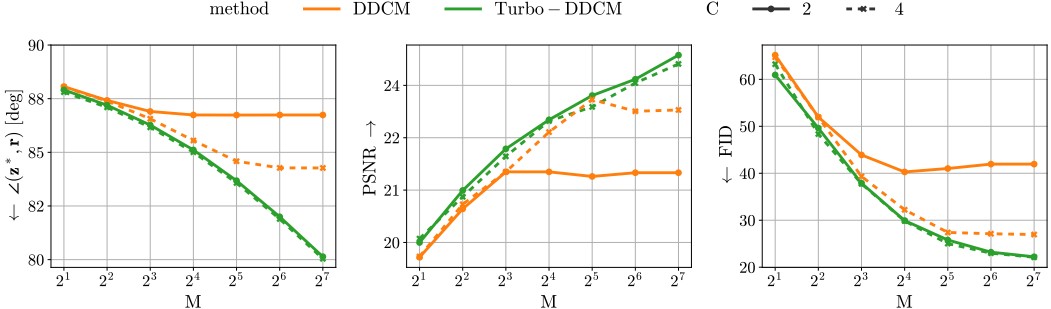

Figure S11: **Effect of hyperparameters on Turbo-DDCM Thresholding and DDCM MP:** Metrics are shown for compression of the Kodak24 ($512 \times 512$) dataset using DDCM and Turbo-DDCM with $T = 100$ and $K = 1024$, while varying $M$ and $C$. Turbo-DDCM consistently approximates the residual at least as well as DDCM MP and continues to improve the approximation with increased $M$, whereas DDCM MP does not improve the approximation beyond a certain point if $C$ is fixed, which results in constant distortion and perceptual performance. Notably, Turbo-DDCM does not scale with $C$; see App. E for details. The $C$ values used differ from our typical value of 1, since DDCM performs MP with $C \geq 2$.

The left plot in Fig. S11 measures $\theta$ during image compression with both methods. It confirms our derivation and shows that Turbo-DDCM consistently provides a solution at least as good as DDCM MP. Moreover, while Turbo-DDCM scales with $M$ and continues to improve the approximation as $M$ increases within the presented range, DDCM MP does not benefit from increasing $M$ alone beyond a certain point; it also requires increasing $C$. This hyperparameter interdependency in DDCM MP can lead to suboptimal configurations and limits fine-grained control over the bitrate, since both hyperparameters must be adjusted simultaneously. In addition, it negatively affects runtime efficiency, as discussed in App. D.

The middle and right plots in Fig. S11 further confirm that the ability to produce a $\mathbf{z}^*$ with a small angle to the residual translates into improved compression performance, both in terms of distortion and perceptual quality. The lack of improvement in the DDCM MP angle is reflected in constant distortion and perceptual quality.

## C.4    INJECTED NOISE DISTRIBUTION

The noise $\mathbf{z}^*$ that we inject into the denoiser is computed using the method described above. In this subsection, we provide a characterization of this random variable.

First, $\mathbf{z}_t^*$ has zero mean, since

$$\mathbb{E}[\mathbf{z}_t^*] = \mathbb{E}\left[\frac{\boldsymbol{C}_t \mathbf{s}_t^*}{\text{std}(\boldsymbol{C}_t \mathbf{s}^*)}\right] \tag{S20}$$

$$= \frac{1}{\text{std}(\boldsymbol{C}_t \mathbf{s}_t^*)} \mathbb{E}[\boldsymbol{C}_t \mathbf{s}^*] \tag{S21}$$

$$= \frac{1}{\text{std}(\boldsymbol{C}_t \mathbf{s}_t^*)} \mathbb{E}\left[\sum_{i=0}^{d} \mathbf{z}_t^{(i)} (\mathbf{s}_t^*)_i\right] \tag{S22}$$

$$= \frac{1}{\text{std}(\boldsymbol{C}_t \mathbf{s}_t^*)} \sum_{i=0}^{d} (\mathbf{s}_t^*)_i \mathbb{E}[\mathbf{z}_t^{(i)}] = 0, \tag{S23}$$

where $\mathbf{z}_t^{(i)}$ is the $i$-th column of $\boldsymbol{C}_t$.

Second, we estimate the covariance matrix of $\mathbf{z}^*$. Figure S12 shows several patches extracted from this matrix for $M \in [10, 150]$. We observe that the entries within these patches are uncorrelated, and each entry exhibits unit standard deviation.

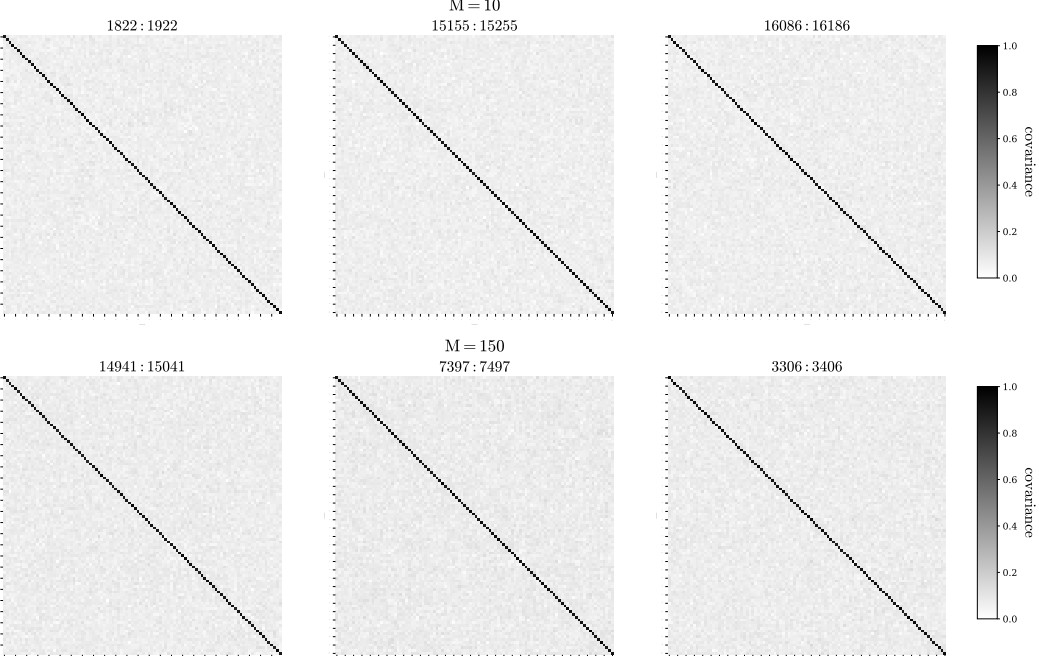

Figure S12: **Patches of the covariance matrix of z\*:** Each subplot shows the empirical covariance of a patch, with the bounds indicated in its subtitle. All entries have unit standard deviation, and different entries appear to be uncorrelated.

# D    THRESHOLDING-BASED STRATEGY EFFICIENCY

This appendix quantifies the computational efficiency of Turbo-DDCM's thresholding strategy relative to DDCM's matching pursuit (MP). As described in Sections 3 and 4, both strategies combine multiple atoms from the codebook to obtain improved solutions to the optimization problems. We provide both a theoretical complexity analysis and empirical results. Importantly, both procedures are performed after each diffusion step (except the last one). As a result, in image compression, the total runtime of these algorithms corresponds to the runtime presented in this appendix multiplied by the number of diffusion steps minus one.

## D.1    THEORETICAL COMPLEXITY COMPARISON

Given a codebook, the MP strategy of DDCM employs a greedy iterative search over $M$ iterations, starting by selecting the atom with the maximum inner product with the residual vector $(x_0 - \hat{x}_{0|t})$. This step requires an exhaustive search over all $K$ atoms, with a computational cost of $\Theta(Kd)$, where $d$ is the latent space dimensionality. In each of the subsequent $M-1$ iterations, the algorithm searches for an additional atom and a quantized coefficient that, together with the current combination, form a convex combination most correlated with the residual. This search is performed via exhaustive search, and after selecting the atom and coefficient, the combination is normalized to unit standard deviation. The process incurs an additional cost of $\Theta((M-1)2^C Kd)$. Therefore, DDCM's MP complexity is

$$\Theta(M2^C Kd). \tag{S24}$$

In contrast, our approach solves an approximate least squares problem under sparsity and quantization constraints (see App. C for the mathematical foundations). We begin by computing the inner products between all atoms and the residual vector, which requires a computational cost of $\Theta(Kd)$. Next, we efficiently select the top-$M$ most correlated atoms, which can be implemented using a heap-based selection algorithm with a cost of $\Theta(K \log M)$. Finally, quantization is performed using the previously computed inner products by finding the nearest quantized coefficient for each of the $M$ selected atoms, with a complexity of $\Theta(2^C M)$. Hence, the overall complexity of our method is

$$\Theta(Kd + K \log M + 2^C M) \approx \Theta(Kd), \tag{S25}$$

since $M \leq K$ and in our typical configurations $M \ll d$ and $C = 1$.

Our algorithm eliminates the dependency on $M$ and $C$ presented in DDCM MP, allowing large values of these parameters to be used without significant additional computational cost. As explained in Section 4, the ability to increase $M$ is critical in our approach for achieving a substantial reduction in the number of diffusion steps, as well as for tuning the bitrate using a single hyperparameter.

## D.2    EMPIRICAL RUNTIME COMPARISON

To quantify the difference between the algorithms and assess the impact of the constants hidden in the $\Theta$ notation, we measure the runtime of both algorithms while varying one parameter at a time among $K$, $M$, and $C$ (with $d$ fixed). Figure S13 confirms our theory and supplies a glimpse for the overall speedup, which passes two orders of magnitude.

Figure S14 shows the acceleration achieved across different combinations of $K$ and $M$. In the configurations that characterize Turbo-DDCM, our thresholding approach achieves two to four orders of magnitude runtime speedup over DDCM approach.

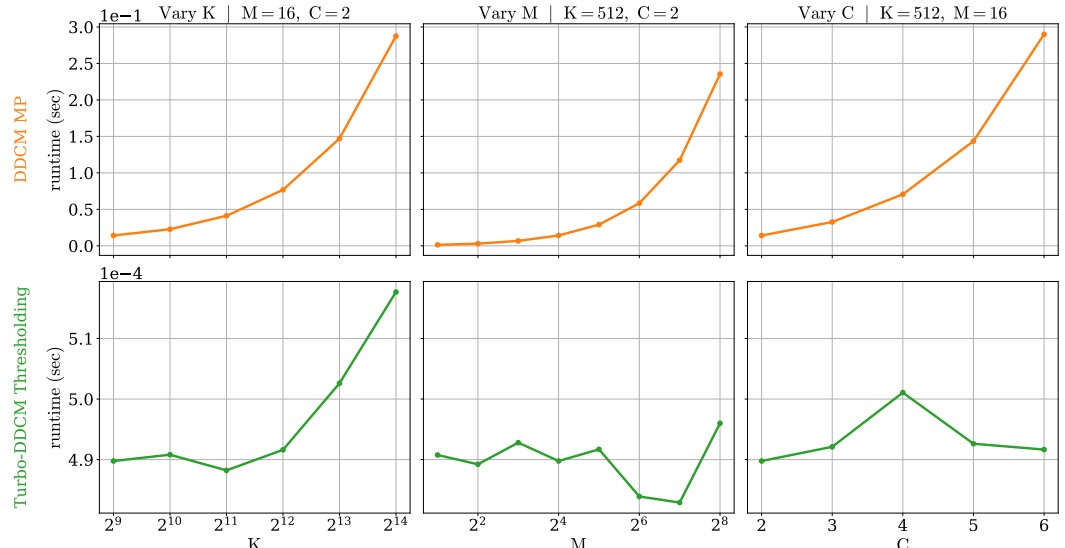

Figure S13: **Impact of hyperparameters on the efficiency of DDCM MP vs. Turbo-DDCM thresholding:** While DDCM MP scales linearly with $K$ and $M$ and exponentially with $C$, our method scales linearly with $K$ and remains nearly constant in $M$ and $C$, achieving over two orders of magnitude speedup.

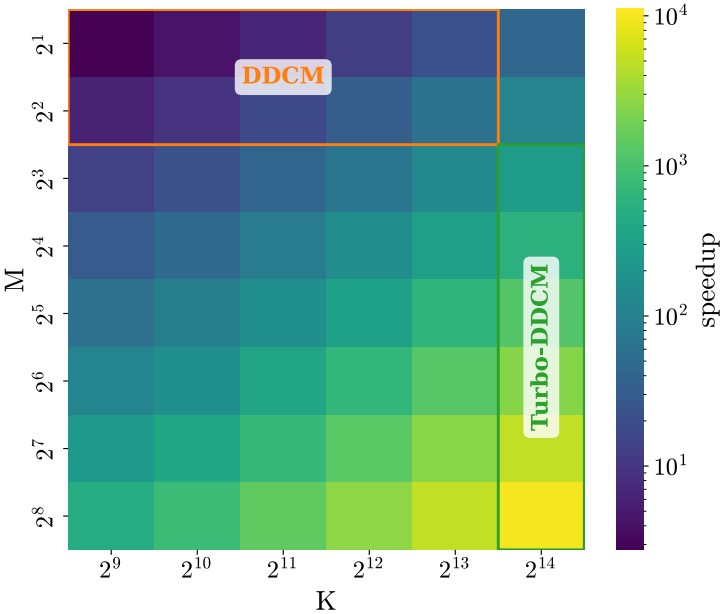

Figure S14: **Relative speedup of Turbo-DDCM thresholding over DDCM MP:** The heatmap shows the speedup using $C = 2$ for both methods, as DDCM MP does not support $C = 1$. The orange box shows DDCM's typical configurations, and the green box shows Turbo-DDCM's. Our method achieves up to four orders of magnitude speedup in the configurations used by Turbo-DDCM, highlighting the difficulty of using DDCM MP in these regions.

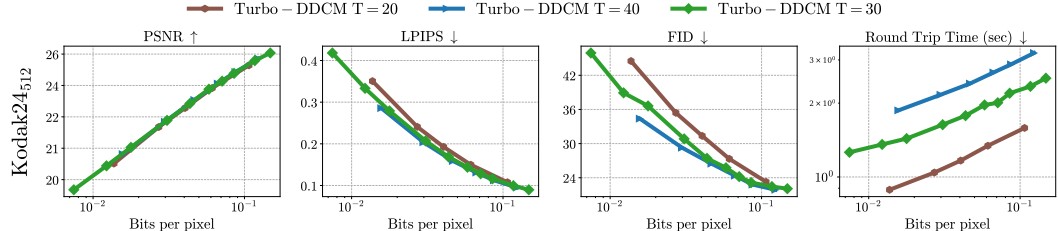

Figure S15: **T impact on Turbo-DDCM:** There is a trade-off between rate-distortion-perception and runtime. $T = 30$ gives minimum runtime without sacrificing performance.

## E  HYPERPARAMETERS

Turbo-DDCM has five hyperparameters:

- $T$: number of diffusion steps.
- $K$: codebook size at each diffusion step.
- $M$: number of selected atoms from the codebook at each step ($M \leq K$).
- $C$: number of bits per quantization coefficient, yielding $2^C$ quantization levels.
- $N$: number of DDIM steps performed at the end without requiring bits.

In this appendix, we discuss the trade-offs between these parameters and shed light on our choice for the base configuration: $T = 30$, $K = 16,384$, $C = 1$, varying $M$ to control bitrate and $N$ ranging from $T - 2$ to 0 depending on the bitrate (see App. A).

We found that $T = 30$ provides the best trade-off between performance and runtime, as presented in Fig. S15. We set $K = 16,384$ because this value is large enough to span a substantial subspace and, in Stable Diffusion 2.1 Base, the entire latent space. At the same time, this $K$ remains small enough to avoid overcompleteness, ensuring a reasonable approximation of orthogonality (see App. B). With $T$ and $K$ fixed, the bitrate is determined primarily by $M$ and $C$.

The hyperparameters $M$ and $C$ directly affect our ability to obtain a high-quality solution to the optimization problem in eq. (9). To understand their marginal impact, we analyze the sources of approximation error in reconstructing the target residual ($\mathbf{x}_0 - \hat{\mathbf{x}}_{t|0}$). The approximation error

$$\|\mathbf{z}^* - (\mathbf{x}_0 - \hat{\mathbf{x}}_{t|0})\|_2^2, \tag{S26}$$

can be decomposed into several components. First, setting constraints aside momentarily, the codebook span may not cover the entire space. Consequently, the best achievable solution is the projection onto the codebook span. If the codebook spans the whole space, this error becomes zero under the assumption that the codebook atoms are linearly independent. Second, we require a sparse solution, so we cannot use the full projection but must find a sparse approximation, which increases the error. Finally, we cannot use this sparse projection directly, but must quantize it, further increasing the approximation error.

Figure S16 illustrates our solutions to the objective with and without the quantization constraint, comparing the sparse projection to the quantized sparse projection with $C = 1$. Notably, $C = 1$, corresponding to quantization coefficients of $\pm 1$, does not introduce traditional quantization levels. Instead, it allows selection of either the codebook noise vector or its opposite, effectively doubling the codebook size. Using $C = 1$, we achieve solutions very close to those of the sparse projection. This behavior can be explained by the fact that the real coefficients are similar in magnitude, making quantization to exact values unnecessary. As a result, we set $C = 1$ to our base configuration.

Finally, we use DDIM steps at low bitrates to improve perceptual quality, but at high bitrates where perceptual quality is already good enough, additional DDIM steps may damage distortion. As a result, $N$ decreases with bitrate. See App. A for exact details on $N$.

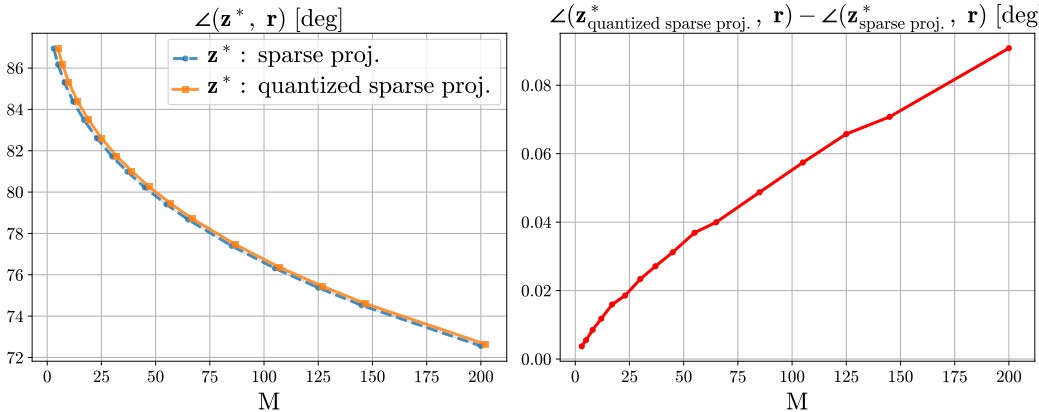

Figure S16: **Quantization impact on objective approximation:** The figure illustrates the difference in approximating our per-diffusion-step objective defined in eq. (9) with and without the quantization constraint. Since we normalize our solutions (eq. (10)), the angle between $\mathbf{z}^*$ and the residual $\mathbf{r}$ is proportional to the norm of their difference. The quantized projection is computed with $C = 1$, and we observe that it yields nearly the same solution as the sparse projection, which corresponds to the solution obtained without the quantization constraint.

## F    BITSTREAM PROTOCOL

As described in Sec. 4, we propose an alternative protocol for bitstream transmission, distinct from that used in DDCM. In this appendix, we primarily show that our protocol achieves relative bit savings exceeding 40% and affects our rate-distortion-perception performance. The BPP formulas for both protocols are given in eq. (8) and eq. (15), with the differences arising from how the selected $M$ atom indices are encoded.

The analyses below are performed for a single diffusion step. Since bits are encoded identically for each step in both protocols, the relative differences similarly apply to the total bit count.

**Relative Bit Saving.**    Let $B_{\text{old}}$ and $B_{\text{new}}$ denote the number of bits required per diffusion step for the original DDCM protocol and our proposed protocol, respectively. Thus,

$$B_{\text{old}} = M\lceil \log_2 K \rceil + MC \quad \text{and} \quad B_{\text{new}} = \left\lceil \log_2 \left( \binom{K}{M} \right) \right\rceil + MC. \tag{S27}$$

The relative saving ratio is therefore

$$\Delta = \frac{B_{\text{old}} - B_{\text{new}}}{B_{\text{old}}}. \tag{S28}$$

Using the Stirling approximation

$$\log_2 \left( \binom{K}{M} \right) \approx M \log_2 \frac{K}{M} + \mathcal{O}(M). \tag{S29}$$

Assuming C is small (see App. E), we can express the saving ratio as

$$\Delta \approx \frac{M \log_2 K - M \log_2 \frac{K}{M}}{M \log_2 K} = \frac{M \log_2 M}{M \log_2 K} = \frac{\log_2 M}{\log_2 K}. \tag{S30}$$

Thus, the relative savings depend logarithmically on the ratio of $M$ to $K$, becoming more significant as $M$ increases relative to $K$. For example, with $K = 16{,}384$ and $M = 100$, which yields a BPP of 0.01 in the new protocol for $T = 30$, $N = 0$, $C = 1$, and an image resolution of $512 \times 512$, the relative reduction is approximately

$$\frac{\log_2 100}{\log_2 16384} \approx 47\%. \tag{S31}$$

**Empirical Evaluation**    We evaluate the relative bit savings by focusing on the transmission of the selected indices, which constitutes the dominant portion of the rate when $C$ is small. As shown in Fig. S17, the empirical results align with the theoretical analysis, demonstrating a substantial reduction in rate (>40%). Moreover, Fig. S18 shows that without the new bit protocol, Turbo-DDCM underperforms in rate-distortion-perception metrics compared to DDCM, whereas with the protocol it outperforms DDCM.

**Lexicographical Combination Encoding and Decoding.**    Each combination $\mathbf{c} = [c_1, \ldots, c_M]$ drawn from $\{0, \ldots, K - 1\}$ can be uniquely represented by its *lexicographical rank*:

$$\text{rank}(\mathbf{c}) = \sum_{i=1}^{M} \sum_{x=c_{i-1}+1}^{c_i-1} \binom{K - x - 1}{M - i}, \quad \text{with } c_0 = -1.$$

This provides a mapping between combinations and integers in $[0, \binom{K}{M})$.

Decoding is performed by iteratively subtracting binomial coefficients:

$$\text{while } \binom{K - x - 1}{M - i - 1} \leq \text{rank}, \quad \text{rank} \leftarrow \text{rank} - \binom{K - x - 1}{M - i - 1}, \ x \leftarrow x + 1,$$

which recovers each element $c_i$ in order. This yields an efficient and reversible mapping between combinations and their lexicographical indices.

**Implementation Details.**    The lexicographic index of each combination is computed efficiently by precomputing binomial coefficients and summing the relevant terms during runtime.

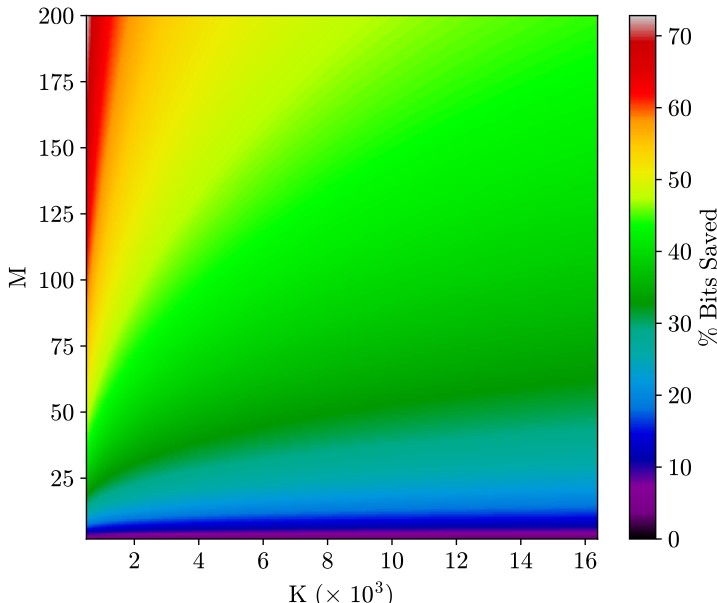

Figure S17: **Relative bit saving:** Percentage of bits saved when transmitting the selected atom indices using our protocol compared to the original DDCM protocol. To illustrate a continuous trend, the plot omits the ceiling operator in the bitrate expressions, which does not affect the asymptotic behavior. For the Turbo-DDCM configuration with $K = 16{,}384$, bitrate savings can exceed 40%.

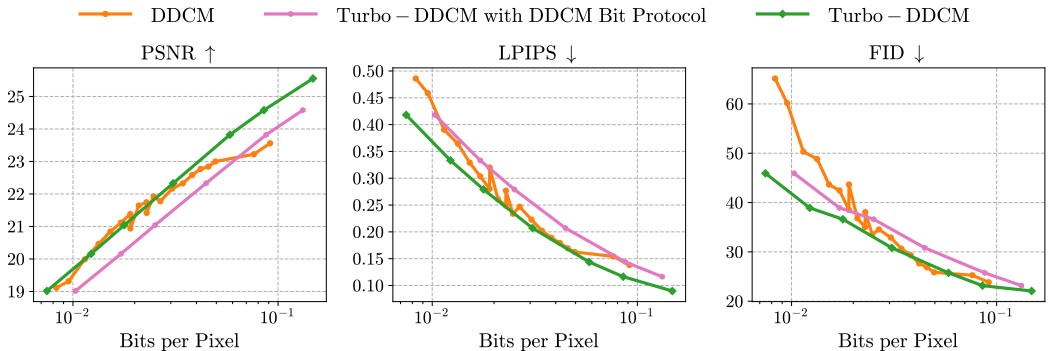

Figure S18: **Impact of bitstream protocol on rate-distortion-perception:** The plot evaluates DDCM, Turbo-DDCM with DDCM protocol, and Turbo-DDCM, which employs our new protocol, on the Kodak24 dataset. While Turbo-DDCM with the original protocol underperforms relative to DDCM, the new protocol allows it to outperform DDCM.

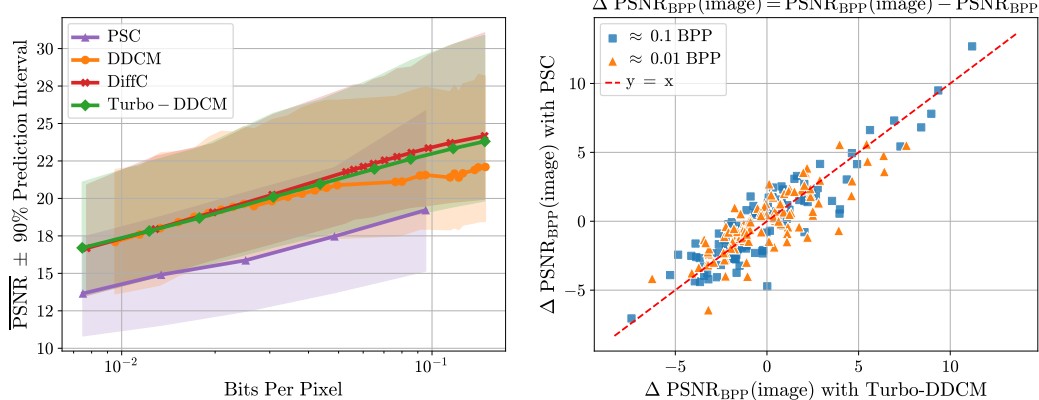

Figure S19: **Distortion variance between images at a fixed bitrate:** All zero-shot methods exhibit high distortion variability across images when evaluated at the same bitrate. In addition, the same images tend to have similar deviations from the mean under both PSC and Turbo-DDCM, despite the fundamental differences between the two methods. Both plots use the DIV2K ($512 \times 512$) dataset.

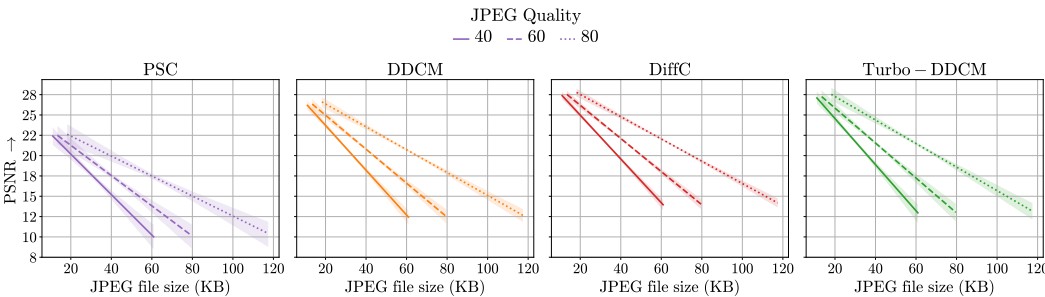

Figure S20: **Correlation between JPEG file size and distortion in zero-shot compression:** Linear regressions are shown between JPEG file size (at different quality levels) and PSNR obtained by each method, for BPPs in the range 0.04-0.07. All methods exhibit a strong inverse correlation. Experiments are conducted on the DIV2K dataset, center-cropped to $512 \times 512$ images.

## G    TURBO-DDCM DISTORTION-CONTROLLED VARIANT

Current diffusion-based image compression algorithms generally take an image and a target bitrate, specified via hyperparameters, as input. However, for a fixed bitrate, the resulting distortion can vary significantly across images. The left plot in Fig. S19 illustrates this for zero-shot diffusion-based methods, showing that PSNR values at a given bitrate can deviate from the mean by more than 2 dB. To improve the predictability of the outputs, it may therefore be preferable to target distortion rather than bitrate.

In this appendix, we propose an efficient method based on Turbo-DDCM, which can also be adapted to other methods, that takes an image and a target distortion as input and predicts the appropriate bitrate to achieve it. We observe that, for a given image, its PSNR deviation from the dataset mean at a fixed bitrate is highly correlated across different methods. For example, the right plot in Fig. S19 demonstrates a strong correlation between PSC and Turbo-DDCM, despite their fundamentally different designs. This observation motivates us to investigate correlations with simple, non-neural methods such as JPEG, which offer ultra-fast compression.

In Fig. S20, we compress images using JPEG at various quality levels and compared the resulting file sizes to PSNR values obtained with zero-shot diffusion-based methods at a specific bitrate. A strong correlation is observed in all quality levels, which may reflect common compression challenges across all methods, such as the difficulty of compressing images with many objects.

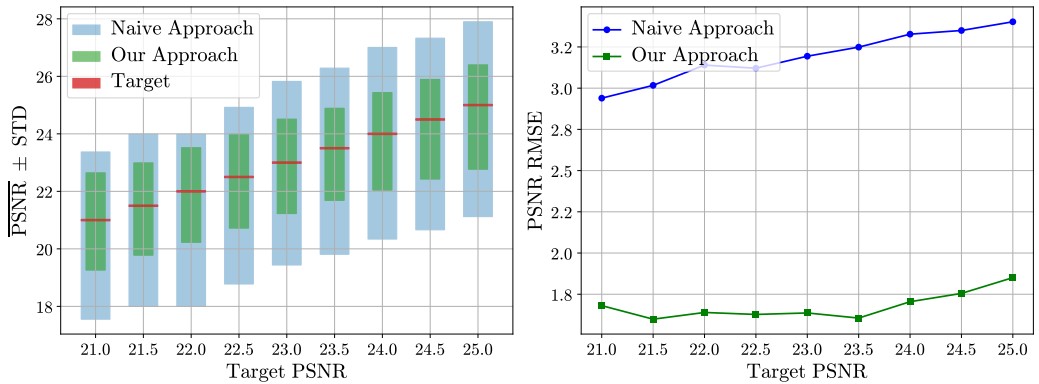

Figure S21: **Distortion control evaluation:** Our method brings the actual distortion much closer to the target, reducing RMSE by over 40% for most target PSNRs. Results in both plots are reported on the test set.

Leveraging this observation, we propose a method for PSNR control with Turbo-DDCM. First, given a training set, compress all images using high-quality JPEG. Next, compress the same images with Turbo-DDCM at various bitrates and learn the correlation between JPEG file size and Turbo-DDCM PSNR for each bitrate via linear regression. To compress a new image to a target PSNR, we first compress it with JPEG and, based on its JPEG file size, predict its Turbo-DDCM PSNR at each bitrate using the learned linear regressions. Finally, we select the bitrate whose predicted PSNR is closest to, but not below, the target. If no such bitrate exists, we choose the highest available bitrate.

To evaluate our approach, we use a large dataset - CLIC2020 (Toderici et al., 2020), with images center-cropped to $512 \times 512$. We first filter out highly difficult images whose JPEG file size at quality 100 exceeds 300 KB, which constitute less than 4% of the dataset. These images are excluded because achieving high PSNR values on them is difficult, almost regardless of the used bitrate. The remaining dataset is split into training and test sets, with 80% of the images used for training and 20% for testing. Both sets are then compressed with JPEG at quality 100. We train a linear regression model for each bitrate on the training set. Finally, for each image in the test set and target PSNR, we determine the bitrate via the above method and evaluate the actual PSNR. The naive approach we compare against selects, for each image, the bitrate whose mean PSNR is closest to the target PSNR. As shown in Fig. S21, our method achieves over a 40% RMSE reduction on the test set across most target PSNR target values compared to the naive approach.

# H DDCM AND TURBO-DDCM VS. DIFFC

DDCM, Turbo-DDCM, and DiffC are all state-of-the-art zero-shot compression methods, achieving comparable compression quality (Fig. 4). While DiffC is based on the reverse-channel-coding (RCC) principle introduced by Theis & Yosri (2022), Turbo-DDCM builds on DDCM (Ohayon et al., 2025), which is not motivated from a solid theoretical foundation. Interestingly, both DDCM-based methods and DiffC optimize the same core objective, yet extend it in fundamentally different ways. In this appendix, we first illustrate DiffC and its underlying motivation, and then highlight its similarities and differences from DDCM and Turbo-DDCM.

## H.1 SHARED CORE OPTIMIZATION OBJECTIVE

All three methods are based on solving optimization problems. The optimization problem of DDCM is presented in Section 3, that of Turbo-DDCM in Section 4, and we now turn to the optimization process of DiffC which is based on RCC. In each diffusion step, DiffC aims to communicate the forward diffusion distribution $q(\mathbf{x}_{t-1} \mid \mathbf{x}_0, \mathbf{x}_t)$ to the decoder, assuming both can sample from the reverse distribution $p_\theta(\mathbf{x}_{t-1} \mid \mathbf{x}_t)$. This communication is performed only for a specific number of initial diffusion steps, which depends on the target bitrate, similar to Turbo-DDCM. The broadcasted information is then used to steer the diffusion process toward the target image via a strategically chosen diffusion noise vector, similar to the approach in Turbo-DDCM.

Vonderfecht & Liu (2025) implement DiffC by applying RCC using the Poisson-Functional-Representation (PFR) approach presented by Theis & Yosri (2022). In their implementation, the PFR algorithm iteratively samples i.i.d. Gaussian noise vectors, selecting the noise $\mathbf{z}_i$ that minimizes $p_\theta(\mathbf{z}_i)/q(\mathbf{z}_i)$. Exploiting the isotropy of both distributions, they simplify this deviation to

$$h_i := \exp\left(-\frac{(\boldsymbol{\mu}_q - \boldsymbol{\mu}_{p_\theta})^T \mathbf{z}_i}{\sigma_{p_\theta}}\right), \tag{S32}$$

where $\boldsymbol{\mu}_q$ and $\boldsymbol{\mu}_{p_\theta}$ are the means of $q$ and $p_\theta$, respectively, and $\sigma_{p_\theta}$ is the standard deviation of $p_\theta$. The iterations for searching the best noise are prioritized, favoring early iterations through an increasing multiplicative weight $w_i = \sum_{j=1}^i \boldsymbol{e}_j$ for the $i$-th iteration, where $\boldsymbol{e}_j \sim \exp(1)$ are i.i.d. exponential random variables. The noise with the minimum weighted score, computed as the product of $w_i$ and $h_i$, replaces the randomly sampled noise in the DDPM reverse process (eq. (2)).

Temporarily ignoring prioritization, DiffC optimization can be formulated as

$$\underset{\mathbf{z}_i}{\arg\min} \exp\left(-\frac{(\boldsymbol{\mu}_q - \boldsymbol{\mu}_{p_\theta})^T \mathbf{z}_i}{\sigma_{p_\theta}}\right) = \underset{\mathbf{z}_i}{\arg\max}(\boldsymbol{\mu}_q - \boldsymbol{\mu}_{p_\theta})^T \mathbf{z}_i. \tag{S33}$$

Placing the values of the means, it simplifies to

$$\underset{\mathbf{z}_i}{\arg\max}(((c_1\mathbf{x}_0 + c_2\mathbf{x}_t) - (c_1\hat{\mathbf{x}}_{0|t} + c_2\mathbf{x}_t))^T \mathbf{z}_i) = \underset{\mathbf{z}_i}{\arg\max}\langle \mathbf{z}_i, \mathbf{x}_0 - \hat{\mathbf{x}}_{0|t}\rangle, \tag{S34}$$

where $c_1$ and $c_2$ are time-dependent constants. This objective is actually the original DDCM objective for the case of $M = 1$, meaning no matching pursuit is performed (eq. (6)). Since $\|\mathbf{x}_0 - \hat{\mathbf{x}}_{0|t}\|_2$ is constant regardless of $\mathbf{z}_i$, and $\|\mathbf{z}_i\|_2$ is approximately constant in high dimensions, eq. (S34) can be written as

$$\underset{\mathbf{z}_i}{\arg\min}\|\mathbf{z}_t - (\mathbf{x}_0 - \hat{\mathbf{x}}_{0|t})\|_2^2, \tag{S35}$$

which is equivalent to Turbo-DDCM's objective for the case of $M = 1$ (eq. (9)). Thus, all three methods steer the diffusion process towards the target image based on the same core objective.

## H.2 DIFFERENCES BETWEEN METHODS

Although these methods share the same core objective, they differ in several key aspects.

**Sampled Noise Prioritization.** As mentioned above, DiffC weights the sampled noises by their sampling iteration number, whereas DDCM and Turbo-DDCM treat all noises equally, without prioritization. While noise prioritization can improve compression on average using variable-length

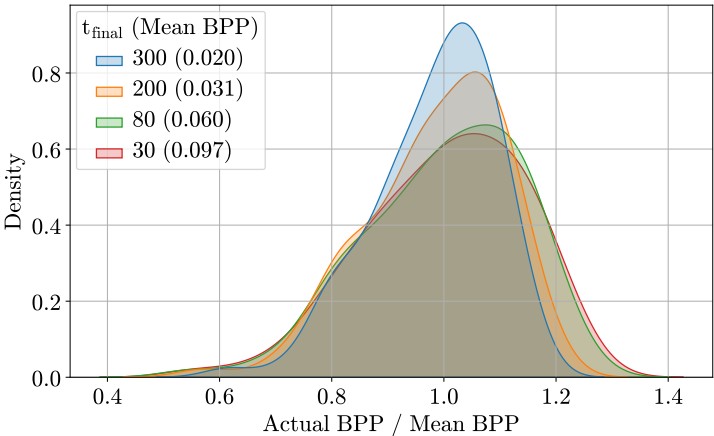

Figure S22: **Actual BPP variance in DiffC across images:** DiffC exhibits variable BPP, which can deviate from the mean by more than 25% under the same BPP target. $t_{final}$ is the final timestep up to which the encoder performs RCC and communicates the result to the decoder. This hyperparameter controls the target BPP.

encoding such as Zipf coding used by DiffC, DDCM and Turbo-DDCM provide a constant and predictable bitrate across images. Figure S22 illustrates that DiffC's bitrate varies significantly between images, with deviations exceeding 25% from the mean, even when using the same target bitrate hyperparameters.

**Increase of the Effective Search Space**  Turbo-DDCM and DiffC employ different strategies to effectively explore a large space of sampled noise vectors. DiffC achieves this by partitioning both $\mathbf{z}_i$ and $\boldsymbol{\mu}_q - \boldsymbol{\mu}_{p_\theta}$ into chunks and independently solving each optimization problem via exhaustive search. If the search is performed over $x$ iterations per chunk across $c$ independent chunks, this effectively explores $x^c$ possible noise maps with only $x \times c$ exhaustive search iterations. In contrast, Turbo-DDCM performs effective multiple-atom selection. This explores $\binom{K}{M} \times 2^{M \times C}$ combinations, where $K$ is the codebook size, $M$ is the number of chosen atoms, and $2^C$ denotes the number of possible non-zero quantization coefficients.

**Search Space Size along Diffusion Steps.**  Supported by theoretical justifications, DiffC progressively increases both the number of chunks and the number of search iterations as the reverse diffusion process advances. In contrast, DDCM and Turbo-DDCM maintain a fixed codebook size and a fixed number of selected atoms across all diffusion steps. Importantly, while DiffC requires a custom CUDA kernel due to performing $2^{16} = 65{,}536$ evaluations per chunk across more than 400 chunks in later diffusion steps, Turbo-DDCM achieves multiple atom selection through an efficient closed-form solution without requiring custom hardware-specific acceleration. The progressively increasing search complexity in DiffC is a major factor behind its growing runtime with bitrate, as shown in Figure 4. As the bitrate increases, the encoder performs more intensive optimization across a greater number of diffusion steps.

### H.3    TURBO-DDCM VS. ACCELERATED DIFFC

Vonderfecht & Liu (2025) mention the option to reduce DiffC runtime by decreasing the number of inference steps. To evaluate both methods under comparable runtimes, we create a faster variant of DiffC using this approach. Since DiffC's runtime varies significantly across bitrates, we reduce the number of diffusion steps to approximately match Turbo-DDCM's runtime at DiffC's minimum runtime. This results in just a few encoding steps for DiffC at low bitrates, increasing up to 10 steps at high bitrates. We refer to this new configuration as *Fast-DiffC*.

Figure S23 shows that Fast-DiffC maintains comparable distortion to the original DiffC across all bitrates, but its perceptual quality is notably inferior to both DiffC and Turbo-DDCM except at the

Figure S23: **Fast-DiffC evaluation:** Fast-DiffC maintains the equivalent distortion to DiffC. However, it has much worse perception. The evaluation uses the Kodak24 ($512 \times 512$) dataset.

highest bitrate. One possible explanation for this phenomenon is that RCC is computationally demanding due to the large number of evaluations required, even with custom CUDA kernel. Thus, Fast-DiffC is forced to perform fewer diffusion steps than Turbo-DDCM, whose inter-step mechanism is more efficient, in order to achieve comparable runtime.

# I  TURBO-DDCM PSEUDO-CODE

---

**Algorithm S1** Turbo-DDCM – Compression

---

1: **Input:** Image to compress $\mathbf{x}_0$, $T$, $K$, $M$, $C$.
2: **Output:** Compressed representation $\mathcal{E}(x_0)$.

3: Calculate $N$.                    ▷ Simple thresholds for $N$ are described in App. A.
4: $\mathcal{E} \leftarrow 0$                    ▷ Initialize the compressed representation
5: $\mathbf{x}_t \leftarrow \text{rand\_normal}(s)$          ▷ $\mathbf{x}_T$ generation. The random seed $s$ is shared with decoder
6: **for** $t = T, \ldots, N + 1$ **do**
7:     $\hat{\mathbf{x}}_{0|t} \leftarrow D(\mathbf{x}_t, t)$                    ▷ Denoiser activation
8:     $\boldsymbol{C}_t \leftarrow \text{rand\_normal}(s + t, \text{K})$.    ▷ codebook sampling with $K$ i.i.d. gaussian latent vectors
9:     $\mathbf{s}_t^* \leftarrow \text{opt}(\boldsymbol{C}_t, \mathbf{x}_0, \hat{\mathbf{x}}_{0|t}, M, C)$       ▷ Solving eq. (9) optimization with eq. (13) closed-form
10:    $\mathbf{z}_t^* \leftarrow \boldsymbol{C}_t s_t^* / \text{std}(\boldsymbol{C}_t s_t^*)$.                    ▷ eq. (10)
11:    calculate $\mathbf{x}_{t-1}$                    ▷ DDPM step
12:    $\text{encoded}_t \leftarrow \text{pack}(\mathbf{s}_t^*)$     ▷ Lexicographic index & quantized coefficients encoding; App. F
13:    $\mathcal{E} \leftarrow \mathcal{E} \| \text{encoded}_t$            ▷ Concatenate step encoding to the compressed representation
14: **end for**
15: **return** $\mathcal{E}(\mathbf{x}_0)$

---

**Algorithm S2** Turbo-DDCM – Decompression

---

1: **Input:** Compressed representation $\mathcal{E}(x_0)$, $T$, $K$, $M$, $C$.
2: **Output:** decompressed image $\tilde{\mathbf{x}}_0$

3: Calculate $N$.                    ▷ Simple thresholds for $N$ are described in App. A
4: $\tilde{\mathbf{x}}_t \leftarrow \text{rand\_normal}(s)$.          ▷ Sampling $\mathbf{x}_T$. Identical to the sampled $\mathbf{x}_T$ in the encoder
5: **for** $t = T, \ldots, N + 1$ **do**
6:     $\hat{\mathbf{x}}_{0|t} \leftarrow D(\mathbf{x}_t, t)$                    ▷ Denoiser activation
7:     $\boldsymbol{C}_t \leftarrow \text{rand\_normal}(s + t, \text{K})$.     ▷ codebook sampling. Identical to the encoder codebook
8:     $\mathbf{s}_t^* \leftarrow \text{unpack}(\mathcal{E})$
9:     $\mathbf{z}_t^* \leftarrow \boldsymbol{C}_t s_t^* / \text{std}(\boldsymbol{C}_t s_t^*)$.                    ▷ eq. (10)
10:    calculate $\tilde{\mathbf{x}}_{t-1}$                    ▷ DDPM step
11: **end for**
12: perform $N + 1$ DDIM steps from $\tilde{\mathbf{x}}_N$
13: **return** $\tilde{\mathbf{x}}_0$

---

See App. A for typical hyperparameters.

## J  LLM USAGE

Large language models (LLMs) were used for minor text polishing and readability improvements.

