# OpenReview forum: "Turbo-DDCM: Fast and Flexible Zero-Shot Diffusion-Based Image Compression"
_ICLR.cc/2026/Conference — ICLR 2026 Poster_

### Official Review · Reviewer_f616 · 2025-10-23

**Soundness:** 4
**Presentation:** 3
**Contribution:** 3
**Rating:** 6
**Confidence:** 4

**Summary:**

This paper addresses the issues of slow speed and high computational resource requirements of DiffC by proposing the Turbo-DDCM method. As a continuation of DiffC methods based on the RCC theory, TurboDDCM well inherits advantages such as zero-shot capability and achieves improvements in speed and performance.

**Strengths:**

1. The motivation is reasonably elaborated and well-supported in the subsequent methods and experiments. The time issue is a key concern in the field of image diffusion research based on diffusion architectures, and the authors have properly proposed a new noise reconstruction method to tackle this.
2. The structure of the paper is clear and well-written.

**Weaknesses:**

1. The authors only conducted time comparisons within the RCC field. It would be valuable to know the speed comparison with diffusion compression methods based on condition introduction and fine-tuning. Specifically, whether RCC-based image compression has time advantages over fine-tuning-diffusion-based image compression, especially given the emergence of many one-step diffusion image compression methods (e.g., StableCodec).
2. Following the first point, for fewer time steps (e.g., 5 steps or even a single step), does the performance of the proposed method degrade significantly?
3. The performance improvement of the final results seems not particularly significant, and the comparisons are insufficient. The baselines selected by the authors appear to be consistent with those of DiffC, but DiffC is a work from ICLR25. It is suggested that the authors include more comparisons with recent works from the past year, such as DiffEIC and StableCodec.
---
ref:

[1] Li Z, Zhou Y, Wei H, et al. Towards extreme image compression with latent feature guidance and diffusion prior[J]. IEEE Transactions on Circuits and Systems for Video Technology, 2024.

[2] Zhang T, Luo X, Li L, et al. StableCodec: Taming One-Step Diffusion for Extreme Image Compression[J]. arXiv preprint arXiv:2506.21977, 2025. (ICCV25 accepted)

**Questions:**

See weaknesses.

---

> ### Author Response · Authors · 2025-11-21
>
> # W1+W3: Runtime Comparison to Trained and Fine-tuned Methods + Comparisons to Additional Methods
> We added runtime comparison to non-RCC-based methods in Fig. 4 and App A.2.
> Even though Turbo-DDCM is zero-shot diffusion-based, it is faster than HiFiC, PerCo (SD), CRDR and DiffEIC. However, there are faster methods than Turbo-DDCM, such as ILLM (non-diffusion-based) and BPG (non-neural) and StableCodec.
>
> We note that our method is zero-shot and offers several advantages over fine-tuned or trained approaches. Beyond avoiding the substantial computational and financial costs associated with training or fine-tuning, zero-shot methods, as discussed in Sec. 1, enable flexible switching of the backbone model. Moreover, in real-world, memory-constrained applications, a single diffusion model can be used for multiple tasks such as image generation, image compression, and image restoration. This effectively provides one versatile tool in place of many task-specific models. Furthermore, to the best of our knowledge, all diffusion-based trained or fine-tuned compression methods, such as StableCodec, are bitrate-specific, which adds a significant overhead in cost and reduces flexibility. In contrast, in Turbo-DDCM the bitrate can be finely controlled via a single hyperparameter over a wide range.
> In addition, our method offers advantages over DiffC that extend beyond its substantially faster runtime: it provides a predictable, fixed-size bitrate across images and does not rely on any custom hardware accelerations.
>
> # W2: $T$ Reduction
> We thank the reviewer for this important point. We added a comprehensive ablation study on the hyperparameter $T$ in App. E. The results indicate that selecting $T=20$ constitutes an optimal balance: further increasing T yields no additional performance gains, whereas reducing T results in a marked degradation in performance.

---

### Official Review · Reviewer_QvxK · 2025-11-05

**Soundness:** 3
**Presentation:** 1
**Contribution:** 2
**Rating:** 2
**Confidence:** 5

**Summary:**

The paper proposes an efficient zero-shot diffusion-based image compression method, which is based on the Denoising Diffusion Codebook Models (DDCMs) compression scheme. The paper modifies DDCM with Turbo-DDCM and introduces two flexible variants of Turbo-DDCM.

**Strengths:**

1. The paper is easy to read.
2. The paper proposes a method which has a faster speed than existing zero-shot diffusion-based image compression methods.

**Weaknesses:**

1. The novelty and contribution are weak. The author just modified the DDCM, but there is no ablation study to analyze the proposed components.
2. The presentation is incomplete. There is a lack of quantitative comparison and ablation study. And there are no tables in the whole paper.

**Questions:**

1. What are the backbones of all zero-shot diffusion-based methods? For a fair comparison of speed, the proposed method should have the same backbone as these methods.
2. Does the author use Turbo-Lora to accelerate Stable Diffusion 2.1?
3. What is the speed of non-diffusion-based methods? The author should also compare with them.
4. How to process higher resolutions?

Please reply to the Weaknesses and Questions. Based on the author's response, I will adjust my rating.

---

> ### Author Response · Authors · 2025-11-21
>
> # W1: Novelty Compared to DDCM
> Although our modifications of DDCM are simple, they have substantial implications. First, Turbo-DDCM achieves the same compression performance as DDCM while being approximately 40$\times$ faster (Fig. 4). Without our proposed Turbo-DDCM, the original DDCM method remains notoriously slow and highly impractical. Furthermore, we introduce two additional variants of Turbo-DDCM (Sec. 6): a distortion-controlled variant which is compatible with the other zero-shot compression frameworks, and a region-of-interest (ROI) solution which allows the user to prioritize the distortion and perceptual quality across different regions of an image. In our revised manuscript, we demonstrate that our ROI solution can be applied to the other RCC-based compression methods (App. A.4). Together, these contributions are a significant advancement beyond the original DDCM formulation.
>
> # W1 + W2: Missing Ablation Studies
> We kindly note that we conducted several ablation studies. First, we ablated how our novel bit protocol and combination mechanism affect the performance of the algorithm in App. F. We showed that using only our new noise combination mechanism speeds DDCM, but deteriorates performance (due to redundancy in the bit protocol, as explained in Sec. 4.2, L321). This shows that our new bit protocol plays an important role in the success of our approach. In addition, we conducted ablation studies on Turbo-DDCM’s hyperparameters in App. E. In particular, we showed that $C=1$ is the best choice for $C$, meaning a finer quantization doesn’t improve performance. In addition, we provided an explanation in the same appendix on why $K=16k$ is a good choice.
> Finally, to round out our ablations, we also add a more comprehensive ablation on $T$ (App. E). The results indicate that selecting $T=20$ constitutes an optimal balance: further increasing $T$ yields no additional performance gains, whereas reducing $T$ results in a significant degradation in performance.
>
> # Q1: Backbone Models
> We kindly note that as described in Sec. 5 (L377), we evaluated all zero-shot methods using the same pre-trained diffusion models: Stable Diffusion 2.1-Base for $512\times512$ images and Stable Diffusion 2.1 for $768\times768$ images.
>
> # Q2: Turbo-Lora
> We didn't use Turbo-Lora to accelerate SD. Rather, we used the standard pre-trained models mentioned above for all zero-shot methods. In other words, our method Turbo-DDCM achieves significant runtime speedups over previous methods without any fine-tuning or other types of manipulation of the pre-trained diffusion model.
>
> # Q3: Non-diffusion-based Methods Speed
> We have added runtime comparison with non-diffusion-based methods in Fig. 4 and App. A.2. Even though our method is zero-shot and diffusion-based, it is faster than HiFiC, PerCo (SD), CRDR and DiffEIC. However, there are faster methods than Turbo-DDCM, such as ILLM (non-diffusion-based) and BPG (non-neural) and StableCodec.
>
> We note that our method is zero-shot and offers several advantages over fine-tuned or trained approaches. Beyond avoiding the substantial computational and financial costs associated with training or fine-tuning, zero-shot methods, as discussed in Sec. 1, enable flexible switching of the backbone model. Moreover, in real-world, memory-constrained applications, a single diffusion model can be used for multiple tasks such as image generation, image compression, and image restoration. This effectively provides one versatile tool in place of many task-specific models. Furthermore, to the best of our knowledge, all diffusion-based trained or fine-tuned compression methods, such as StableCodec, are bitrate-specific, which adds a significant overhead in cost and reduces flexibility. In contrast, in Turbo-DDCM the bitrate can be finely controlled via a single hyperparameter over a wide range.
>
>
> # Q4: Higher Resolutions Processing
> Similarly to DDCM, the ability of Turbo-DDCM to generalize for different image resolutions depends solely on the ability of the underlying pre-trained diffusion model to handle such resolutions. To adjust to a new resolution, one simply needs to substitute a pre-trained diffusion model of the desired resolution and adjust the dimensionality of the codebook to match that model. If a shared-seed random generator is not available, the codebooks can be pre-sampled at the maximal supported resolution and the image resolution can then be transmitted, and smaller codebooks can be dynamically sliced from the pre-sampled set as needed.

---

### Official Review · Reviewer_NNvr · 2025-11-05

**Soundness:** 4
**Presentation:** 4
**Contribution:** 4
**Rating:** 8
**Confidence:** 5

**Summary:**

Turbo-DDCM proposes a new approach to zero-shot diffusion-based compression. In the original Denoising Diffusion Codebook Models (DDCM) approach, at each denoising timestep, the compression algorithm selects the next noise vector to add to the image from a codebook of K noise vectors, y selecting the one which moves the noisy image closest to the target image. In Turbo-DDCM, the authors instead encode a noise vector by selecting a subset of M of those K random vectors, and taking a linear combination of them, with coefficients either -1 or 1. This effectively allows the authors to select from a combinatorially larger set of noise vectors, getting closer to the target image with each step. With this improved guidance, they can steer the diffusion model to produce the target image in many fewer steps. They empirically show that this method allows for zero-shot image compression using diffusion models which is multiple times faster than prior approaches.

**Strengths:**

This is a clever idea for improving the computational efficiency of noise-vector selection in zero-shot DDPM-based compression. It clearly works quite well.

**Weaknesses:**

I consider all of these zero-shot methods (DiffC, DDCM, Turbo-DDCM) to be modifications of Theis et al's original DiffC proposal. They all, to various degrees, trade off theoretical elegance/rigor for real-world usability. In the original 2022 algorithm, there's a precisely defined relationship between the diffusion model's log probability of an image and the number of bits needed to compress that image to a certain noise level. Turbo-DDCM achieves state-of-the-art real-world performance, but trades away these theoretical guarantees.

**Questions:**

The other available zero-shot diffusion compression methods (DDCM, and DiffC) have hyperparameters for speeding up their runtime by trading off against their rate/distortion performance. Primarily this means the number of denoising steps performed. It would be very interesting to see how these methods fare as you decrease the number of denoising steps to match Turbo-DDCM.

It's not obvious to me that the z_t* vectors selected by turbo-DDCM should yield x_{t-1} vectors which are in the same distribution that the diffusion model was trained on? Like the diffusion model was trained to expect specific linear combinations of real images + gaussian noise, at different ratios, for each time step. But when you generate z_t* according to this fancy process, does it still have the same statistics that random noise does, from the perspective of the diffusion model? Like you are selecting these vectors to be correlated with the target image, so I would naively expect them to have different statistics from random noise. But I'm not confident in this assessment.

---

> ### Author Response · Authors · 2025-11-21
>
> # Q1: Decreasing Runtime in Other Methods
> We kindly note that we discussed such a comparison with DiffC in App. H. We showed that when the runtime of DiffC is reduced (by lowering the number of diffusion steps) to match that of Turbo-DDCM, DiffC is limited to only a few diffusion steps. This is due to DiffC’s inter-step processing mechanism, which incurs a non-negligible computational overhead even when it employs a custom CUDA kernel. Consequently, its perceptual quality degrades significantly and becomes notably worse than that of Turbo-DDCM, for the same runtime.
>
> Regarding DDCM, it is not feasible to significantly reduce their $T$. Achieving a similar target bitrate would require substantially increasing the other hyperparameters ($K,M,C$), which results in extremely slow runtimes (as discussed in Sec. 3 and presented theoretically and empirically in App. D). In contrast, a key advantage of our method is that the asymptotical runtime complexity of the combination mechanism is independent of $M$ and $C$, allowing these values to be increased arbitrarily without incurring any additional computational burden.
>
> # Q2: $\mathbf{z}_t^*$ Distribution
> This is a good point. To get a feeling of how $\mathbf{z}_t^*$ looks like, we added a proof in App. C.4 that it has zero mean, as well as empirical evidence that it has a unit standard deviation and uncorrelated entries.

---

### Official Review · Reviewer_4uEv · 2025-11-08

**Soundness:** 3
**Presentation:** 2
**Contribution:** 2
**Rating:** 4
**Confidence:** 4

**Summary:**

This paper proposes a fast and flexible zero-shot diffusion-based image compression method (i.e., Turbo-DDCM), which reduces the number of required denoising operations and maintains the performance with an improved encoding protocol. Moreover, Turbo-DDCM presents a priority-aware variant that prioritizes regions of interest and a distortion-controlled variant that compresses an image based on a target PSNR. Experiments are performed on Kodak and DIV2K datasets to investigate the effectiveness of the proposed method.

**Strengths:**

Turbo-DDCM has a competitive performance with recent methods in terms of the rate-distortion-perception tradeoff, and it achieves up to an order of magnitude speedup over existing methods.

**Weaknesses:**

1.	The experimental section lacks comprehensive quantitative results, such as BD-rate or BD-PSNR.
2.	The workflow description of the proposed method is not concise enough.
3.	Although the proposed method is faster than the comparison algorithm, it does not offer a performance advantage.
4.	The paper does not analyze the advantages of diffusion-based image compression methods compared to other types of image compression models, such as GAN, CNN, and RNN.
5.	No results tables are provided for better indicating its superiorities over other methods.

**Questions:**

1.	What is the meaning of the symbol C in Eq. (8)?
2.	The author should provide a detailed explanation of the encoding and decoding process in Figure 2.
3.	What is the meaning of log2(K M) in Eq. (14)?
4.	It is difficult to distinguish the differences among the decoded images in the first row of Figure 1.
5.	In Figure 4, when comparing with other methods, the authors should also provide the results on the DIV2K dataset.
6.	The proposed method should be compared with GAN-based algorithms.
7.	When comparing computational complexity, the authors should also provide GPU memory usage.
8.	The author should analyze the impact of the values of hyperparameters T, K, and C on performance.

---

> ### Author Response · Authors · 2025-11-21
>
> # W1+W5: BD-rate + Lack of Tables
> We thank the reviewer for suggesting to also report the BD-rate and providing exact tables. We added a BD-rate analysis table in App. A.3. The results support the observation that Turbo-DDCM outperforms PSC in distortion, matches DDCM, and is marginally below DiffC. In addition, we added in App. A tables for the results from Fig 4.
>
> # W3: Lack of Performance Advantage
> Please note that our goal was to at least preserve the compression performance of DDCM, while reducing its runtime significantly. We indeed achieve this goal, as Turbo-DDCM is orders of magnitude faster than DDCM (Fig. 4). Moreover, we preserved favorable characteristics of DDCM in terms of compression (which do not exist in DiffC), like not being dependent on custom hardware accelerations, having constant and predictable bitrate between images and having nearly constant runtime across bitrates. The fixed and predictable bitrate can be a major advantage in real world settings. For example, transmitting large sets of images over a fixed-bandwidth network without sudden data spikes or delays.
> We would like to emphasize that our contribution is not only speeding up DDCM. In particular, we also introduce two novel variants in Sec. 6. The first is for ROI-aware compression and the second for distortion-controlled compression, which were not shown before to work with zero-shot diffusion-based compression methods. In the revised manuscript (in App. A.4) we present that our ROI method is applicable for other RCC-based zero-shot methods.
>
> # W4+Q6: Comparison to Non-diffusion-based Methods
> We thank the reviewer for pointing this out. We added a comparison to HiFiC, a GAN-based method with a CNN architecture (Fig 4. and App. A). HiFiC offers lower distortion than Turbo-DDCM but worse perceptual quality. In addition, HiFiC is slower than Turbo-DDCM and it lacks a flexible bitrate range.  Furthermore, we would like to note that Fig. 4 and App. A included a comparison with ILLM, which is also GAN-based, as well as BPG, a non-neural method.
>
> # Q1: C in Eq. (8)
> We are sorry for the confusion. $C$ is the number of bits that are required to communicate a single quantized coefficient (which equals $\log_2$ of the set size of quantized coefficients). We clarified this in the manuscript.
>
> # Q2: Pseudocode
> We thank the reviewer for this suggestion. We added a pseudo code of our compression and decompression algorithms in App. I.
>
> # Q3: log2(K M) in Eq. (14)
> We are sorry for this typo. The equation should read $\log_2 \left(\binom{K}{M} \right)$, i.e., the binomial coefficient. We address this issue in the revised manuscript.
>
> # Q4: Differences Among the First-row Decoded Images in Figure 1
> Indeed, the visual quality of all methods is similar, which  demonstrates that our method produces images on par with previous state-of-the-art methods (e.g., DiffC and DDCM) quality-wise, with a significantly faster runtime (denoted at the top of each column).
>
> # Q5: Comparison on DIV2K Dataset with Non-zero-shot Methods
> We kindly note that such a comparison was presented in App. A. We moved it to Fig. 4 in the revised manuscript.
>
> # Q7: GPU Memory Considerations
> We thank the reviewer for this important suggestion. We added a subsection (App. A.5), showing that Turbo-DDCM requires a peak overhead of just 500MB GPU memory (on top of the 2.5GB required by the diffusion model) for compressing images of size $512\times512$, corresponding to a 20% increase. This memory is freed and reused at each diffusion step. Moreover, Turbo-DDCM can be implemented even more efficiently to consume a negligible memory overhead by sampling one codebook atom at a time, computing its score ($(u_t)_i$ in Eq. 13), retaining its index only if the score is in the top M scores, and reusing the memory to compute scores for subsequent codebook atoms.
>
> # Q8: Impact of The Hyperparameters T,K,C on Performance
> We kindly note that the impact of the hyperparameters $K$ and $C$ was discussed in App. E. We have added a comprehensive ablation on the hyperparameter $T$. The results indicate that our selection of $T=20$ constitutes an optimal balance: further increasing $T$ yields no additional performance gains, whereas reducing $T$ results in a significant degradation in performance.

---

### Official Review · Reviewer_4dvB · 2025-11-09

**Soundness:** 2
**Presentation:** 3
**Contribution:** 3
**Rating:** 4
**Confidence:** 3

**Summary:**

The paper introduces a zero-shot algorithm based on a pre-trained diffusion model to achieve image compression at extremely low bit rates. In essence, this method represents an efficient enhancement of the DDCM approach, transforming the original MP method into MULTI-ATOM and proposing a new bit protocol to further reduce the redundancy in the bitstream generated by the MULTI-ATOM method. The paper presents a plethora of experiments and evidence to validate the effectiveness and reliability of the algorithm. The experiments demonstrate that this approach achieves excellent performance metrics in zero-shot diffusion-based schemes.

**Strengths:**

1. The paper is well-written, with clear expression and concise yet precise explanations of the motivation and methodology.
2. The theoretical explanations are solid, providing a high level of theoretical underpinning for the approach.
3. The experiments are comprehensive, comparing all zero-shot diffusion-based image compression baselines and achieving satisfactory performance even with a several-fold acceleration

**Weaknesses:**

1. Some modules lack sufficient explanations, especially those borrowed from other articles. Providing necessary explanations can help readers better understand the operational mechanisms of the entire algorithm.
2. The lack of striking innovativeness in the algorithm may be noted, as the entire solution builds upon the pipeline of DDCM and does not surpass DiffC in terms of performance.
3. I believe confining the baselines to the zero-shot domain is inappropriate. While zero-shot algorithms show promise and are worth researching in diffusion-based image compression schemes, all zero-shot algorithms fundamentally sacrifice rapid inference to reduce or eliminate training costs, whereas compression tasks are sensitive to inference latency. Recently, some solutions based on few-step pre-trained diffusion models [1,2] have achieved outstanding performance with minimal latency. Overall, I suggest that the method should be compared against some non-zero-shot diffusion baselines or explore the potential application of this algorithm in fine-tuning diffusion priors, which could enhance the practical contributions and persuasiveness of the paper.

[1] [TCSVT] RDEIC: Accelerating Diffusion-Based Extreme Image Compression with Relay Residual Diffusion

[2] [ICCV 2025] StableCodec: Taming One-Step Diffusion for Extreme Image Compression

**Questions:**

1. The article's overview of the overall encoding and decoding process is somewhat vague. It would be beneficial to include a pseudocode  to provide readers with a clearer understanding of the algorithm's encoding and decoding processes.
2. The use of ROI in the article for flexible bitrate allocation is commendable. However, the paper lacks details on the application of this technology. For instance, is ROI specific to this algorithm only? Can it be integrated with DDCM or other zero-shot schemes? What is the specific process of implementing ROI? Does ROI introduce additional inference latency? These aspects should be further elaborated.
3. The article showcases the Round Trip Time. What does this specifically refer to? In a compression algorithm, it's essential to separately compare encoding and decoding times.
4. The article employs a 512 central crop for the Kodak dataset, which is relatively uncommon in the compression field where full-size Kodak testing is more prevalent. Is there a specific reason for this approach? Is it related to high GPU memory usage by the algorithm?
5. Testing FID on the Kodak dataset may not be appropriate as Kodak comprises only 24 images, and even when divided into 64-pixel patches, reliable FID metrics may not be guaranteed.
6. CLIC20 is a commonly used dataset in image compression. It would enhance the algorithm's reliability to include CLIC20 in the main experiments.

---

> ### Author Response · Authors · 2025-11-21
>
> # W1 + Q1: Improved Explanations and Pseudocode
> Thanks for pointing this out. We improved the readability of our paper by adding more explanations and revising some of the text, especially for the modules originating from DDCM (see paragraph 2 of Sec.1 and paragraph 2 of Sec. 3.2 in the updated manuscript). In addition, we have added a pseudo-code of our method in App. I. We hope that these revisions make our manuscript easier to understand.
>
> # W2 + Q2: Innovativeness + ROI variant
> Although our modifications of DDCM are simple, they enable Turbo-DDCM to run orders of magnitude faster than DDCM, while maintaining comparable rate-distortion-perception performance. Moreover, although our method is on par with DiffC in terms of rate-distortion-perception, Turbo-DDCM remains up to an order of magnitude faster with nearly constant runtime across bitrates, despite the fact that DiffC benefits from custom CUDA kernel accelerations. Lastly, as opposed to DiffC, Turbo-DDCM offers a fixed and predictable bitrate across images, a property that is highly desirable in practical compression settings. Together, these advantages make Turbo-DDCM a significantly more practical compression engine.
> Furthermore, as we showed in App. H, when the number of steps in DiffC is reduced to match the roundtrip time of Turbo-DDCM, DiffC’s performance drops substantially below ours. Thus, while DiffC achieves slightly better rate-distortion-perception under its default configuration, it does so at the expense of speed and practicality.
> We would like to emphasize that our contribution is not only speeding up DDCM. In particular, we also introduce two novel variants in Sec. 6. The first is for ROI compression and the second for distortion-controlled compression, which were not shown before with zero-shot diffusion-based compression methods. We thank the reviewer for commending our ROI variant. As detailed in Sec. 6 (L491), the ROI-variant is obtained by point-wise multiplying the residual vector with the mask $w$ (Eq. 16) at each diffusion step. The resulting masked vector is then treated as the residual in the subsequent optimization. and  Indeed, our method for ROI compression is general for all RCC-based methods. Based on the reviewer's suggestion, we added new results in App. A, where we implement our ROI compression variant for DDCM and DiffC. Regarding latency, our approach for ROI compression needs only pointwise-multiplication of the residual vector with the given prioritization map $w$ (eq. 16) in each step, and therefore, as mentioned in Sec. 5 (L493), our ROI algorithm introduces a negligible runtime overhead. Thus, here as well, Turbo-DDCM remains the fastest approach.
>
> # W3: Comparison to Non-zero-shot Methods
> We kindly note that we had comparisons to non-zero-shot methods in the initial manuscript, including non-diffusion-based methods and even non-neural methods in Fig. 4 and App. A. It can be noticed that Turbo-DDCM has superior perceptual quality than almost all other methods. This comes on top of the advantages of zero-shot methods. For example, avoiding the substantial computational cost of training while allowing flexible backbone switching, as discussed in Sec. 1. Moreover, in memory-constrained real-world applications, a single diffusion model can serve multiple tasks, such as generation, compression, and restoration, providing one versatile tool in place of many specialized models. The disadvantage is indeed inference runtime and this is what we try to solve. And actually, we are faster than some non-zero-shot methods, like DiffEIC and HiFiC, as presented in Fig. 4. Furthermore, to the best of our knowledge, all diffusion-based trained or fine-tuned compression methods, such as StableCodec, are bitrate-specific, which adds a significant overhead in cost and reduces flexibility. In contrast, in Turbo-DDCM the bitrate can be finely controlled via a single hyperparameter over a wide range.
>
>
> # Q3: Roundtrip Time
> Roundtrip time, as we define in Sec.1, is the total amount of time it takes to encode an image and then to decode it. Following the reviewer’s question, we have added in App. A separate timing measurement for encoding and decoding . The results show that our speed-up over DDCM is both in the encoding and the decoding phases. We thank the reviewer for this suggestion.

---

> > ### Author Response · Authors · 2025-11-21
> > **Cont.**
> >
> > # Q4: Kodak Resize + GPU Memory
> > The resolution of our method and its performance on different resolutions is directly related to the resolution of the base model used. For example, Stable Diffusion 2.1 is meant for an image resolution of $768\times768$, and we find empirically it produces distorted images when the image size is less than the intended resolution. Similarly,  Stable Diffusion 2.1-Base is intended for $512\times512$ images, and we empirically find a worse performance on larger images. Therefore, since the images in Kodak24 are $512\times768$ (or $768\times512$), to fairly compare the compression scheme performance without biasing the results we cropped the images to be aligned to SD 2.1-Base. Specifically, this was not done because of some memory limitation. This is a shared limitation of all the zero-shot methods. Larger datasets, such as DIV2K, were cropped to the higher possible resolution of $768\times768$, as reported in Fig. 4.
> > Nevertheless, we conduct in App. A a comparison of memory usage, which shows that in terms of peak memory, Turbo-DDCM requires only 500MB on top of the SD 2.1 (2.5GB) for 512x512 images, corresponding to a 20% increase. Moreover, Turbo-DDCM can be implemented even more efficiently to consume a negligible memory overhead by sampling one codebook atom at a time, computing its score ($(u_t)_i$ in Eq. 13), retaining its index only if the score is in the top M scores, and reusing the memory to compute scores for subsequent codebook atoms.
> >
> > # Q5: FID Patches
> > We used the FID implementation of *neuralcompression* library, which takes 50% overlapping patches of the images, which results in 113 patches for each $512\times512$ image. This results in 2712 patches for all of Kodak24.
> >
> > # Q6: CLIC2020
> > We thank the reviewer for suggesting to validate our results on CLIC2020, beyond the existing experiments on DIV2K and Kodak24. We followed this advice and added comparisons to CLIC2020 in App. A, which exhibit the same trends as on the Kodak24 and DIV2K datasets.

---

### Comment · Area_Chair_ocby · 2025-11-25

Dear Reviewers,

Thank you for your time and effort in reviewing submissions for ICLR 2026. As we begin the author-reviewer discussion process, we kindly remind you to submit your responses to the author rebuttals by **December 2**.

Your engagement in this discussion phase is crucial to ensuring a fair and thorough evaluation of each submission.

### **Action Required**
- Carefully consider the authors’ rebuttal and any additional evidence they provide.
- Update your review (if applicable) to reflect your revised perspective.
- Discuss with the authors if further details are required

Your AC

---

### Author Response · Authors · 2025-11-30
**Revision Summary**

We sincerely thank the reviewers for their thoughtful comments. In response, we have added further clarifications and conducted additional experiments. We provide a summary of the main changes below:
1) **Turbo-DDCM Pseudo-Code** - added in App. I.
2) **ROI Variant** - demonstrated that our ROI compression approach used in Turbo-DDCM (Sec. 6) can also be applied to the other RCC-based compression methods (such as DDCM and DiffC). The results included in App. A.
3) **Additional Non-zero-shot Methods** - added comparisons to HiFiC, StableCodec and DiffEIC.
4) **Runtime Comparison to Non-zero-shot Methods** - added in Fig. 4 and App. A.
5) **Evaluation on CLIC2020** - added in App. A.
6) **GPU Memory Considerations** - added GPU memory analysis in App. A.
7) **Ablation Study on the Hyperparameter T** - added to App. E.

We hope this summary will help streamline your review of the revised manuscript.

---

### Meta-Review · Area_Chair_vVjm · 2026-01-06

**Summary:**

This paper proposes Turbo-DDCM, a substantially accelerated variant of the zero-shot diffusion-based image compression method DDCM, achieving approximately 40$\times$ speedup while maintaining comparable performance.

The submission initially received mixed scores (8, 6, 4, 4, 2), reflecting differing views on the paper’s contribution. Reviewer QvxK (score 2) and 4dvB (score 4) raised concerns regarding limited novelty, arguing that Turbo-DDCM is based on relatively simple modifications of DDCM. However, authors argue that the combination mechanism and redesigned bit protocol are essential to achieving the reported speedup without degrading compression quality. The ablation studies presented in the rebuttal helped prove this point. Other reviewers viewed these changes as non-trivial.

Several reviewers requested broader comparisons with fine-tuned or training-based compression methods, including StableCodec. While Turbo-DDCM does not outperform StableCodec in efficiency, the authors convincingly argue that their method offers distinct practical advantages, including:
- a zero-shot formulation without training or fine-tuning costs,
- flexible bitrate control via a single hyperparameter,
- compatibility with arbitrary pre-trained diffusion backbones, and
- predictable runtime and bitrate across images.

However, the current experiments are conducted primarily using Stable Diffusion 2.1, which is not strong enough to show the flexibility. AC therefore suggested demonstrating the method with additional diffusion models (including distilled diffusion models, or flow-matching-based models) to further validate its generalizability. These experiments, if they contain an HR generative model, will also be helpful to respond to reviewer QvxK's question about how to process HR images. Besides, experiments on processing images with different resolutions would be useful for showing the flexibility.


Most remaining concerns pertained to clarity, missing evaluations, and experimental completeness, all of which were adequately addressed in the rebuttal through added pseudocode, runtime and memory analysis, ROI generalization experiments, and results on additional datasets.

Overall, despite some disagreement regarding the degree of novelty, the paper demonstrates a clear and meaningful practical contribution to making zero-shot diffusion-based image compression feasible. Given the strengthened experimental support and satisfactory resolution of reviewer concerns, AC recommends acceptance but expects the authors to improve their paper according to the suggestions.

**Reviewer Concerns:**

## Addressed Concerns

**Clarity of the method and encoding/decoding Pipeline**
Several reviewers (4dvB, 4uEv, QvxK) noted that the original manuscript lacked a clear description of the encoding and decoding process, with ambiguous notation and insufficient explanation of several components. The authors added detailed explanations, corrected notation, and provided full pseudocode in the appendix, substantially improving readability and technical clarity.

**Missing quantitative evaluations and tables**
Reviewers requested more comprehensive quantitative results, including BD-rate / BD-PSNR analyses, explicit result tables, runtime breakdowns, GPU memory usage, and evaluations on additional datasets such as CLIC2020. The rebuttal added BD-rate tables, separate encoding/decoding timing, GPU memory analysis, and new experiments on CLIC2020, confirming consistent trends across datasets.

**Comparison with additional baselines**
Multiple reviewers asked for comparisons with non-zero-shot and non-diffusion-based methods, including GAN-based and trained diffusion approaches (e.g., HiFiC, DiffEIC, StableCodec). The revised manuscript includes these comparisons and clarifies the tradeoffs between zero-shot flexibility and trained-model performance.

**ROI variant generality**
Reviewers requested clarification on whether the proposed ROI compression is specific to Turbo-DDCM and whether it introduces additional computational cost. The authors demonstrated that the ROI mechanism generalizes to other RCC-based methods (DDCM, DiffC) and incurs negligible runtime overhead.

**Hyperparameter sensitivity and ablations**
Concerns were raised regarding the effect of key hyperparameters and the lack of ablation studies. The rebuttal added extensive ablation studies and clarified the role and impact of these parameters.

**Degree of novelty relative to DDCM**
Two Reviewer QvxK (score 2) and 4dvB (score 4) raised concerns regarding limited novelty, arguing that Turbo-DDCM is based on relatively simple modifications of DDCM. However, authors argue that the combination mechanism and redesigned bit protocol are essential to achieving the reported speedup without degrading compression quality. The ablation studies presented in the rebuttal helped prove this point. Other reviewers viewed these changes as non-trivial.

## Outstanding Concerns
**Advantage over Fine-tuning-based Methods**
AC suggest to show more evidence about the flexibility of the zero-shot-based method.

**Reviewer Scores:**

Reviewer NNvr (8 $\rightarrow$ 8)
This reviewer was already strongly positive, highlighting the technical idea, empirical effectiveness, and practical impact of Turbo-DDCM. The rebuttal addressed their technical questions regarding runtime comparisons and different noise statistics.

Reviewer f616 (6 $\rightarrow$ 6/8)
This reviewer requested broader comparisons with trained and fine-tuned diffusion methods and clarification on performance under reduced diffusion steps. These points were addressed through additional comparisons, runtime analysis, and ablation studies.

Reviewer 4dvB (4 $\rightarrow$ 6/8)
This reviewer raised concerns about limited novelty, lack of non-zero-shot comparisons, missing pseudocode, dataset coverage (CLIC2020), runtime breakdowns, and ROI generalizability. All of these issues were addressed in the rebuttal with new experiments, clarifications, and analyses.

Reviewer 4uEv (4 $\rightarrow$ 6)
This reviewer emphasized missing quantitative tables, BD-rate analysis, GPU memory usage, hyperparameter ablations, and broader baseline comparisons. These requests were comprehensively addressed in the revised manuscript. Remaining concerns primarily relate to the perceived balance between speed and compression performance.

Reviewer QvxK (2 $\rightarrow$ 4/6)
This reviewer strongly questioned the novelty and completeness of the experimental evaluation. The rebuttal addressed their technical questions, presented ablations, clarified backbone usage, expanded baseline comparisons, and strengthened the experimental section. Regarding novelty, the authors respond that the $40\times$ speed improvement is non-trivial and that the two proposed components are new. From AC's perspective, it is convincing.

---

### Decision · Program_Chairs · 2026-01-26

Accept (Poster)